# Equalized Generative Treatment: Matching $f$-Divergences for Fairness in Generative Models

Alexandre Vérine [1]   Rafael Pinot [2]   Florian Le Bronnec [3]

## Abstract

Fairness is a crucial concern for generative models, which not only reflect but can also amplify societal and cultural biases. Existing fairness notions for generative models are largely adapted from classification and focus on balancing the probability of generating samples from each sensitive group. We show that such criteria are brittle, as they can be met even when different sensitive groups are modeled with widely varying quality. To address this limitation, we introduce a new fairness definition for generative models, termed as *equalized generative treatment* (EGT), which requires comparable generation quality across all sensitive groups, with quality measured via a reference $f$-divergence. We further analyze the trade-offs induced by EGT, demonstrating that enforcing fairness constraints necessarily couples the overall model quality to that of the most challenging group to approximate. This indicates that a simple yet efficient Min–Max fine-tuning method should be able to balance $f$-divergences across sensitive groups to satisfy EGT. We validate this theoretical insight through a set of experiments on both image and text generation tasks. We demonstrate that Min–Max methods consistently achieve fairer outcomes compared to other approaches from the literature, while maintaining competitive overall performance for both tasks.

## 1. Introduction

The rise of generative models has revolutionized a wide range of domains, including natural language processing, computer vision, and scientific discovery (Hu et al., 2026).

As these systems become ever more ubiquitous, ensuring their *trustworthiness* has emerged as a critical challenge (Kucharavy et al., 2024). Trustworthiness is inherently multidimensional, encompassing many challenges such as robustness to perturbations (Carlini et al., 2024), protection of user privacy (Carlini et al., 2023; 2022), and fairness (Choi et al., 2020; Zameshina et al., 2022). Among them, concerns about fairness are particularly pressing, as generative models not only reflect but also shape the ways information and cultural artifacts are created, distributed, and consumed. Left unchecked, these systems risk amplifying and existing societal biases and stereotypes. Despite growing interest, most approaches to fairness in generative models fail to provide definitions that reflect the unique characteristics of generative modeling. Existing metrics are often adapted directly from the literature on fair classification, focusing on recalibrating generative odds (i.e., the probability of being generated) across sensitive groups (Choi et al., 2020; Zameshina et al., 2022; Teo et al., 2023b;a). However, such criteria fall short in capturing nuanced disparities on how different subpopulations are modeled.

This apparent lack of formalization in addressing fairness issues may stem from the broader challenge of assessing the quality of generative models, which remains an open problem. At its core, generative modeling seeks to approximate a target data distribution using a (parametrized) model distribution. A common strategy involves minimizing a statistical $f$-divergence, where the choice of $f$ defines the objective and, consequently, the properties of the trained model (Goodfellow et al., 2014; Nowozin et al., 2016; Grover et al., 2018). At evaluation, however, the process becomes more complex due to the coexistence of a multitude of competing metrics. While $f$-divergences can off course be employed for evaluation (e.g., for computing the precision and recall of the model (Verine et al., 2023)) other metrics such as the FID (Heusel et al., 2017), MAUVE (Pillutla et al., 2021), or self-BLEU (Zhu et al., 2018)), are also frequently used in practice. These alternatives, however, often represent ad hoc choices with limited or no theoretical grounding, offering little insight to analysts. To establish a clear and general framework for analyzing fairness in generative models, we adopt $f$-divergences. This choice ensures consistency with existing training objectives while providing a principled

[1]DI ENS, École normale supérieure, Université PSL, CNRS, Paris, France [2]Sorbonne Université, Université Paris Cité, CNRS, Laboratoire de Probabilités, Statistique et Modélisation, LPSM, Paris, France [3]Institute of Science Tokyo, Tokyo, Japan. Correspondence to: Alexandre Vérine <alexandre.verine@ens.fr>.

*Proceedings of the 43$^{rd}$ International Conference on Machine Learning*, Seoul, South Korea. PMLR 306, 2026. Copyright 2026 by the author(s).

foundation for our fairness analysis, as detailed further in Section 2.1. In this context, our contributions are as follows.

Our central contribution is to move the fairness question from *"who gets generated"* to *"how well each group is generated"*. Existing proportion-based criteria can ensure that a model generates the right number of samples from each sensitive group, but they do not control whether these groups receive comparable fidelity and diversity. Equalized Generative Treatment (EGT) is introduced precisely to fill this gap: it requires the group-conditional approximation errors, measured through $f$-divergences, to be comparable across groups.

**Brittleness of existing fairness criterion.** We first show in Theorem 3.1 that existing notions of fairness can yield models that, while well-calibrated in terms of generative odds, can be arbitrarily unbalanced in terms of $f$-divergence between the target and trained conditional distribution for each sensitive group. This means that even when satisfying these definitions, models can generate some groups with much higher quality and diversity than some others. This identifies the precise failure mode that EGT is designed to resolve: matching group proportions alone can hide low-quality generations concentrated in one group. This outcome may not seem surprising, as existing methods were not designed to address this issue directly. Nevertheless, it underscores how the current lack of formalization can result in approaches that provide a false sense of fairness and provide a formal proof of the brittleness of existing definitions. We validate this theoretical result with numerical experiments for image generation on the FFHQ dataset (Karras et al., 2020) or for text on the Wikipedia Biographies dataset (Bronnec et al., 2024). In each case, satisfying existing definitions do not prevent the model from treating each sensitive group differently in terms of generation quality.

**A new definition matching $f$-divergences.** Based on this observation, we introduce a new definition of fairness called *equalized generative treatment* (EGT), which relies on simultaneously controlling the $f$-divergence between the target and trained distributions for each sensitive group. In Theorem 4.3, we demonstrate that applying this definition promotes the minimization of the highest $f$-divergence among sensitive groups. This naturally indicates that Min–Max fine-tuning methods are good candidates to target our new fairness criterion. We compare these methods to existing fairness solutions in the literature, evaluating performance on image generation using the FFHQ dataset (Karras et al., 2020) and on text generation using the Wikipedia Biographies dataset (Bronnec et al., 2024). While some methods may coincidentally improve EGT for specific model/task combinations, Min–Max schemes consistently outperform or match prior work on this criterion across all settings we consider. In short, we show that Min–Max fine-tuning meth-

ods provide a stable and theoretically grounded solution to the fairness problem in generative models.

## 2. Background & Related Works

Let $\mathcal{X} \subset \mathbb{R}^d$, endowed with the euclidean norm $\|\cdot\|$. We denote $\mathcal{P}_\lambda(\mathcal{X})$ the set of probability distribution on $\mathcal{X}$ that are absolutely continuous w.r.t. the Lebesgue measure $\lambda$. For any $P \in \mathcal{P}_\lambda(\mathcal{X})$, its probability density function (pdf) is denoted by $p = \frac{dP}{d\lambda}$. Finally, we denote $\Delta(\mathcal{A})$ the probability simplex over a finite set $\mathcal{A}$, and $\mathbb{1}_E(x)$ the indicator function of the event $x \in E$, for any $E \subset \mathcal{X}$.

### 2.1. $f$-Divergences in Generative Modeling

Learning a generative model can be formalized as approximating a target distribution $P \in \mathcal{P}_\lambda(\mathcal{X})$ with a model distribution $Q$, where $Q$ belongs to an admissible family $\mathcal{Q} \subset \mathcal{P}_\lambda(\mathcal{X})$. To measure the discrepancy between $P$ and $Q$, we use the general class of $f$-divergences, defined below.

**Definition 2.1.** Let $f : (0, +\infty) \to (-\infty, +\infty]$ be a convex, lower semi-continuous function with $f(1) = 0$. For any $P, Q \in \mathcal{P}_\lambda(\mathcal{X})$, the $f$-divergence between the distributions $P$ and $Q$ is defined as

$$\mathcal{D}_f(P\|Q) = \int_{\text{SUPP}(Q)} f\left(\frac{p(x)}{q(x)}\right) q(x)\, d\lambda(x) + \zeta(\text{SUPP}(Q)),$$

where $p$ and $q$ are the pdfs of $P$ and $Q$, $\text{SUPP}(Q) = \{x \in \mathcal{X} \mid q(x) > 0\}$ denotes the support of $Q$, $\zeta(\text{SUPP}(Q)) = \bar{f}(\infty)\, P(\mathcal{X}\backslash\text{SUPP}(Q))$, and $\bar{f}(\infty) = \lim_{t\to+\infty} f(t)/t$. Furthermore, this definition adopts the conventions $f(0) = \lim_{t\to 0^+} f(t)$ and $0 \times \infty = 0$ to avoid the above being ill-defined when $\bar{f}(\infty) = \infty$ and $P(\mathcal{X} \backslash \text{SUPP}(Q)) = 0$.

The choice of $f$ in Definition 2.1 specifies the divergence we aim to compute. For instance, the Kullback-Leibler divergence can be obtained by setting $f(t) = t \log t$. This divergence is usually minimized by likelihood-based methods used in LLMs (Grattafiori et al., 2024), in Normalizing Flows (Rezende & Mohamed, 2015) or in some score-based diffusion models (Song et al., 2021). On the other hand, Generative Adversarial Networks (Goodfellow et al., 2014) typically optimize objective functions related to the Jensen-Shannon divergence, defined for $f(t) = t \log t - (t+1) \log(t+1)/2$. Additionally, several methods have also been proposed to minimize the Total Variation in GANs (Um & Suh, 2021) or LLMs (Ji et al., 2023). More generally, recent work has proposed modular frameworks that allow targeting a variety of $f$-divergences, enabling practitioners to tailor the objective to the specific context and application image modeling (Nowozin et al., 2016; Grover et al., 2018; Cai et al., 2020; Verine et al., 2023; Xu et al., 2025) and more recently for LLMs (Wang et al., 2023; Sun

& Schaar, 2024; Go et al., 2023; Verine et al., 2025).

**Evaluation.** In practice, there exists many metrics to evaluate the performance of a generative model, each with their own strengths and weaknesses (Borji, 2023). Among these metrics, a prominent example is the family of metrics that separately measure quality and diversity, most notably *precision* and *recall* for generative models (Kynkäänniemi et al., 2019) refined by Kim et al. (2023a). Given two distributions $P$ and $Q$, these metrics are defined as

- $\mathrm{Precision}(Q\|P) = Q\big(\mathrm{SUPP}(P)\big),$

- $\mathrm{Recall}(Q\|P) = P\big(\mathrm{SUPP}(Q)\big).$

Precision and recall can be interpreted as a particular instance of $f$-divergences as demonstrated by (Simon et al., 2019; Verine, 2024). Furthermore, extensions and generalizations of these metrics building on $f$-divergences have been investigated in several recent works (Sajjadi et al., 2018; Djolonga et al., 2020; Pillutla et al., 2021). Furthermore, these two metrics are often combined within a Precision/Recall divergence which is simply defined as the sum of the $\mathrm{Precision}(Q\|P) + \mathrm{Recall}(Q\|P)$. Note that, by additivity of $f$-divergences, Precision/Recall is also a $f$-divergence (as a sum of $f$-divergences). More explicitly, under the generalized precision–recall formulation of Sajjadi et al. (2018), precision and recall can be written as

$$\mathrm{Precision}(Q\|P) = 1 - \mathcal{D}_{f_\mathrm{P}}(P\|Q),$$
$$\mathrm{Recall}(Q\|P) = 1 - \mathcal{D}_{f_\mathrm{R}}(P\|Q),$$

for suitable choices of $f_\mathrm{P}$ and $f_\mathrm{R}$ (Verine et al., 2023). Therefore, precision and recall themselves are metrics to be maximized, while the corresponding $f$-divergences are minimized. In contrast, when we later report $\delta$-Precision or $\delta$-Recall, the symbol $\delta$ denotes a disparity across sensitive groups and smaller values are better.

## 2.2. Fairness in Generative Modeling

In the context of fairness, we extend this problem formulation by introducing a set of sensitive attributes $\mathcal{A}$ together with an oracle function $\psi : \mathcal{X} \to \mathcal{A}$ that maps each instance $x \in \mathcal{X}$ to its corresponding sensitive attribute. We assume $\psi$ is defined over the entire space, i.e., $\mathrm{dom}(\psi) = \mathcal{X}$. This induces a partition of $\mathcal{X}$ into disjoint subsets $\mathcal{X}_a = \{x \in \mathcal{X} \mid \psi(x) = a\}$ for each $a \in \mathcal{A}$. For any distribution $P \in \mathcal{P}_\lambda(\mathcal{X})$ with pdf $p$, we can then express $P$ as the mixture

$$P = \sum_{a \in \mathcal{A}} \pi_a^P P_a \quad \text{with} \quad p_a(x) = \frac{p(x)}{\pi_a^P} \mathbb{1}_{\mathcal{X}_a}(x) \quad \forall x \in \mathcal{X},$$

where $\pi_a^P = P(\mathcal{X}_a)$ denotes the proportion of the population associated with attribute $a$, and $P_a$ is the conditional

distribution restricted to $\mathcal{X}_a$. Thus, the vector $(\pi_a^P)_{a \in \mathcal{A}}$ lies in the simplex $\Delta(\mathcal{A})$, and $P$ can be interpreted as a mixture of attribute-conditional distributions. We call a distribution $P \in \mathcal{P}_\lambda(\mathcal{X})$ non-trivial if $\pi_a^P > 0$ for all $a \in \mathcal{A}$. In the following, we will always assume the target distribution we consider is non-trivial, since fairness considerations would otherwise be meaningless.

**Existing fairness criteria.** Most existing approaches to fairness in generative modeling focus on the proportions of sensitive attributes in the generated distribution. This perspective was popularized by Hutchinson & Mitchell (2019), who characterized fairness as the equal representation of a chosen sensitive attribute among generated samples. To the best of our knowledge, no standard terminology has been established in the literature for such proportion-based criteria. We therefore introduce the following nomenclature: *equalized generative odds* (EGO), and *matching generative odds* (MGO). The distinction between these two notions reflects different goals: EGO enforces uniform representation across groups, whereas MGO requires the generative model $Q$ to reproduce the proportions of the target distribution $P$.

**Definition 2.2** (Equalized Generative Odds (EGO)). Let $Q \in \mathcal{P}_\lambda(\mathcal{X})$ and $\delta > 0$. $Q$ is said to satisfy $\delta$-*equalized generative odds* if

$$\left|\pi_a^Q - \pi_{a'}^Q\right| \le \delta, \quad \text{for all } a, a' \in \mathcal{A}.$$

When $\delta = 0$, we say that $Q$ satisfies *equalized generative odds*.

**Definition 2.3** (Matching Generative Odds (MGO)). Let $P, Q \in \mathcal{P}_\lambda(\mathcal{X})$ and $\delta > 0$. $P$ and $Q$ are said to satisfy $\delta$-*matching generative odds* if

$$\left|\pi_a^Q - \pi_a^P\right| \le \delta, \quad \text{for all } a \in \mathcal{A}.$$

When $\delta = 0$, we say that $P$ and $Q$ satisfy *matching generative odds*, meaning that the group proportions under $Q$ exactly match those of $P$.

**Existing methods.** Most approaches to fairness in generative modeling have focused on enforcing group proportions, either through EGO or MGO. A variety of methods have been developed with this goal in mind. For instance, Choi et al. (2020) reweight the training distribution, enforcing EGO when the data is unbiased and approximating MGO otherwise. This method has since been widely adopted and extended in the literature (Yazdani-Jahromi et al., 2024; Yan et al., 2022; Kim et al., 2024) and adapted to diffusion models by Kim et al. (2023b). Other works modify the generation process itself, thereby directly controlling the proportion. For instance, Frankel & Vendrow (2018); Humayun et al. (2021); Tan et al. (2021)regularize the latent space to reduce deviations from MGO, while Zameshina et al.

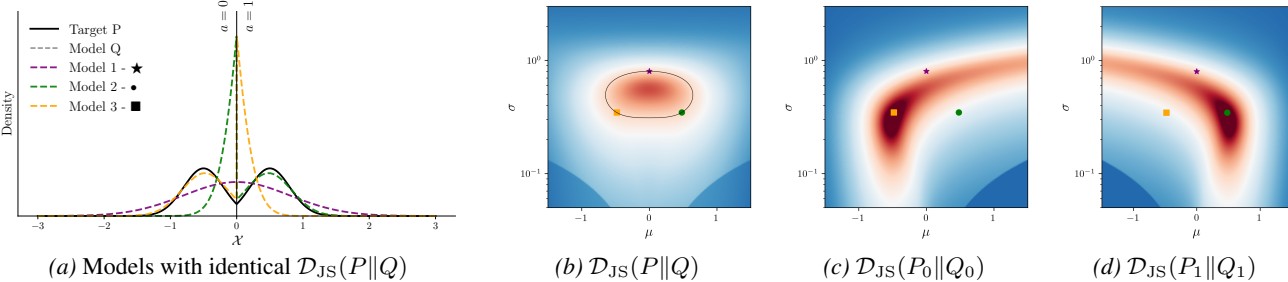

*(a) Models with identical $\mathcal{D}_{\mathrm{JS}}(P\|Q)$*     *(b) $\mathcal{D}_{\mathrm{JS}}(P\|Q)$*     *(c) $\mathcal{D}_{\mathrm{JS}}(P_0\|Q_0)$*     *(d) $\mathcal{D}_{\mathrm{JS}}(P_1\|Q_1)$*

*Figure 1.* Jensen–Shannon divergence between a target distribution $P$ and rescaled Gaussian models $Q = \mathcal{N}(\mu, \sigma^2)$. (1a) Models with the same global divergence $\mathcal{D}_{\mathrm{JS}}(P\|Q)$ can still differ greatly. Model 1, Model 2, and Model 3 are represented respectively by a purple ★, a green ●, and an orange ■. (1b) Level set for $\mathcal{D}_{\mathrm{JS}} = 1$, with selected models marked by their corresponding markers. (1c)–(1d) Conditional divergences for the two groups; models on the same level set may yield highly unbalanced conditional divergences. In panels (1b)–(1d), darker red indicates larger divergence values, while darker blue indicates smaller divergence values.

(2022) apply rejection sampling based on predicted sensitive attributes, which can be tuned to enforce either EGO or MGO. Specifically for diffusion, Parihar et al. (2024) focus on guidance to ensure EGO. On the evaluation side, Teo et al. (2023a) proposed classifier-based metrics designed to assess how well generative models satisfy EGO or MGO, but again the focus remained solely on proportions. Only very recently has work begun to go beyond this perspective. Other work such as Mayer et al. (2024) and Um & Suh (2021) introduced an evaluation based on the FID between majority and minority groups, although this was limited to the case of synthetic data generation or GANS specific approach. In summary, while most prior work has focused on improving or evaluating fairness through proportions (EGO or MGO), little attention has been given to local distributional discrepancies, which can be more effectively captured by $f$-divergences for both evaluation and training.

## 3. On the Brittleness of Existing Definitions

In Section 2.2, we discussed how most existing approaches to fairness in generative modeling are framed in terms of *matching generative odds* (MGO) or *equalized generative odds* (EGO). These notions capture fairness only at the level of group proportions, without accounting for the local behaviors (namely, how accurately the conditionals $(P_a)_{a \in \mathcal{A}}$ are reproduced). In this section, we argue that such a focus on proportions is inherently fragile. In particular, we show that even when MGO and EGO are perfectly satisfied, substantial discrepancies can (and does) remain in how different sensitive groups are modeled. We first illustrate this limitation through a simple example, then formalize it in a general theoretical result, and finally validate it empirically on state-of-the-art models.

### 3.1. Theoretical Brittleness of Matching and Equalized Generative Odds

As a warm-up, consider the setting illustrated in Figure 1, where $\mathcal{X} = \mathbb{R}$, $\mathcal{A} = \{0, 1\}$, and the oracle function is $\psi = \mathbb{1}_{\mathbb{R}_+}$. The target distribution is defined as $P = \frac{1}{2}P_0 + \frac{1}{2}P_1$, where $P_0$ and $P_1$ are truncated Gaussian distributions with respective means $\pm 0.5$ and standard deviation $0.3$. As model family $\mathcal{Q}$, we consider the set of univariate Gaussian distributions $\{\mathcal{N}(\mu, \sigma^2) \mid (\mu, \sigma) \in \mathbb{R} \times \mathbb{R}_+\}$ that we rescale on each side of the origin to enforce MGO and EGO. To examine the potential disparities arising under these criteria, we focus on the Jensen Shannon divergence $\mathcal{D}_{\mathrm{JS}}$. For this divergence, we study the level set $\{Q \in \mathcal{Q} \mid \mathcal{D}_{\mathrm{JS}}(P\|Q) = \epsilon\}$, for a fixed level $\epsilon = 1$. This value is used only to visualize one representative nonzero level set under our numerical convention; the same qualitative phenomenon appears for other nonzero levels $\epsilon$.

Figure 1b shows that this level set allows for several pairs of admissible parameters $(\mu, \sigma) \in \mathbb{R} \times \mathbb{R}_+$, with three representative solutions highlighted by distinct markers. The purple ★, green ●, and orange ■ correspond to Model 1, Model 2, and Model 3, respectively. On the one hand, for one selected model in Figure 1b, the conditional divergences $\mathcal{D}_{\mathrm{JS}}(P_0\|Q_0)$ and $\mathcal{D}_{\mathrm{JS}}(P_1\|Q_1)$ are both small and almost identical. On the other hand, for the other selected models, a clear imbalance appears. Indeed, in both cases, one of the conditional divergences is as low as $0.02$, while the other reaches $1.98$. This illustrative example shows that even when MGO and EGO are perfectly satisfied, the model can still exhibit arbitrarily poor fidelity for one of the sensitive groups.

**More general result.** The brittleness observed above is not a mere artifact of our toy example, but rather a general phenomenon. Even when $\mathcal{Q}$ is allowed to range over all distributions in $\mathcal{P}_\lambda(\mathcal{X})$ that satisfy EGO and MGO with $P$, and even when the global $f$-divergence $\mathcal{D}_f(P\|Q)$ is

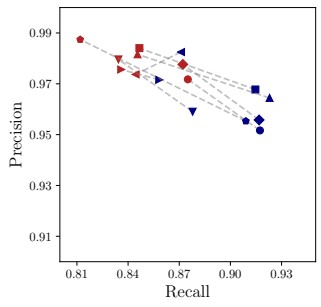
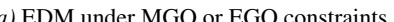
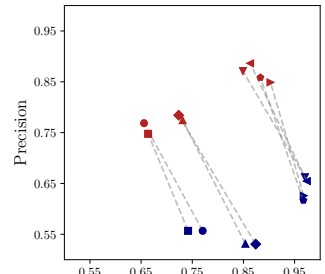

*(a)* EDM under MGO or EGO constraints.

*(b)* LLaMA-3.2-Chat & Gemma-3 under MGO constraints.

*Figure 2.* Precision and recall for EDM (VE and VP) on FFHQ (2a) and for LLaMA-3.2-Chat (1B and 3B) and Gemma-3 4B (pretrained and instruction-tuned) on Wikipedia Biographies (2b) under two settings: pretrained and conditional. At the sampling stage, rejection sampling is used to enforce either MGO or EGO. Each sensitive group is color-coded (red or blue), and the points corresponding to the subgroups for a given model are connected by a dashed line. We observe that significant discrepancies in precision and recall persist across groups, demonstrating the brittleness of proportion-based definitions.

constrained to be arbitrarily small (but non-zero), it remains possible to design situations in which one group incurs an arbitrarily larger conditional divergence than the others. To make this precise, by analogy with the warm-up above, we introduce the *$f$-divergence level set* around $P$ as

$$\mathcal{S}_{\mathcal{D}_f}(P,\epsilon) \;=\; \big\{ Q \in \mathcal{P}_\lambda\left(\mathcal{X}\right) \;\big|\; \mathcal{D}_f(P\|Q) = \epsilon \big\}.$$

By working within $\mathcal{P}_\lambda\left(\mathcal{X}\right)$, we impose no restriction on model expressivity, thereby going beyond the specificity of our toy example. In this setting, the phenomenon of imbalanced conditional divergences across attributes can be formalized as follows.

**Theorem 3.1.** *Let $P \in \mathcal{P}_\lambda\left(\mathcal{X}\right)$ be a non-trivial target distribution satisfying EGO, and let $f$ be a continuous function such that $\mathcal{D}_f$ defines an $f$-divergence. For any $\epsilon \in (0, f(0) + \bar{f}(+\infty))$ and any $\gamma \in (0, \epsilon)$, there exists $Q^\gamma \in \mathcal{S}_{\mathcal{D}_f}(P,\epsilon)$ such that $Q^\gamma$ satisfies MGO with $P$, but for which there exists $\bar{a} \in \mathcal{A}$ with*

$$\mathcal{D}_f(P_{\bar{a}}\|Q_{\bar{a}}^\gamma) \;\geq\; \mathcal{D}_f(P_a\|Q_a^\gamma) + \gamma, \qquad \forall a \in \mathcal{A} \setminus \{\bar{a}\}.$$

In other words, even when both EGO and MGO are met, it is always possible to build a model performing arbitrarily worse on one group than on the others. This establishes that global fairness criteria based solely on proportions provide no guarantee of conditional quality for the groups.

### 3.2. Observing the Brittleness in Practice

The brittleness highlighted above is not only theoretical: it also manifests in practice with off-the-shelf generative models. To demonstrate this, we run experiments on competitive diffusion models (images) and large language models (text). Experimental details are provided below, with additional information in Appendix B.

**Datasets and sensitive attributes.** Two modalities are considered: images and text. For image generation, we consider FFHQ (Karras et al., 2021) and focus on gender as the sensitive attribute (44% male, 56% female). For text generation, experiments rely on a dataset derived from Wikipedia Biographies (Bronnec et al., 2024), again using gender as the sensitive attribute (approximately 75% male, and 25% female respectively).

**Methods.** An unconditional baseline (pretrained by the original authors) is compared to a conditional baseline that explicitly conditions on the sensitive attribute, enabling direct control of group proportions. For a fair comparison, rejection sampling is applied to the unconditional model so that generated samples match the desired sensitive-group proportions (Zameshina et al., 2022). MGO or EGO is enforced for the image task, while only MGO is enforced for the text task due to computational constraints.

**Models.** Image generation relies on Elucidated Diffusion Models (EDM) (Karras et al., 2022) with both variance-preserving (VP) and variance-exploding (VE) parameterizations. The unconditional baseline uses the authors' pretrained checkpoints, while the conditional model is obtained by fine-tuning on FFHQ using class-conditional embeddings. Text generation uses LLaMA-3.2-Chat (Grattafiori et al., 2024) (1B and 3B) and Gemma-3 4B (Team et al., 2025) in both pretrained (pt) and instruction-tuned (it) variants. Conditional generations are produced by prompting the base models with the sensitive attribute.

**Evaluation metrics and oracle functions.** Group-wise performance is assessed via precision and recall per sensitive group using Topological Precision and Recall (TopP&R) (Kim et al., 2023a). This procedure requires an embedding model and an oracle function $\psi$ defining the sensitive groups. For FFHQ, the oracle is a fine-tuned DINOv2 ViT (Oquab et al., 2024) gender classifier, and its embeddings are used for TopP&R. For Wikipedia Biographies, a keyword-based oracle is enough to perfectly classify the attributes, and we

used Qwen3-4B text-embeddings (Zhang et al., 2025).

**Observed disparities.** Figure 2 reports precision and recall by sensitive group. Despite enforcing MGO (and EGO when applicable), substantial disparities persist. For diffusion models on FFHQ, group gaps reach up to 3.2% in precision and 7.7% in recall. For LLaMA-3.2 and Gemma-3 on Wikipedia Biographies, disparities are larger, reaching up to 25.37% in precision and 15.06% in recall. In most cases, precision is higher for the minority group (female), whereas recall is higher for the majority group (male), suggesting a systematic trade-off across groups. Overall, these results empirically validate the brittleness of MGO/EGO: satisfying proportion-based criteria does not prevent large group-wise differences in conditional quality.

# 4. Equalized Generative Treatment (EGT)

The analysis in Section 3 shows that fairness criteria based solely on group proportions, such as MGO and EGO, are inherently brittle. To address this limitation, we introduce a simple yet fundamental fairness criterion, called *equalized generative treatment* (EGT). We then show that EGT forces the global divergence to account for the worst group-conditional approximation error, motivating Min–Max training as a practical candidate rather than as a direct consequence of a tight characterization.

## 4.1. Definition and First Property

We now introduce the notion of *equalized generative treatment* (EGT). This criterion provides a stronger, more fine-grained notion of fairness by explicitly linking generative quality to each subgroup using a reference $f$-divergence.

**Definition 4.1** (Equalized Generative Treatment (EGT)). Let $P, Q \in \mathcal{P}_\lambda(\mathcal{X})$ and let $f$ be such that $\mathcal{D}_f$ defines an $f$-divergence. For any $\delta > 0$, we say that $Q$ and $P$ satisfy $\delta$-*equalized generative treatment* ($\delta$-EGT) w.r.t. $\mathcal{D}_f$ if for all $a, a' \in \mathcal{A}$ one has

$$\left| \mathcal{D}_f(P_a \| Q_a) - \mathcal{D}_f(P_{a'} \| Q_{a'}) \right| \leq \delta.$$

When $\delta = 0$, we say that $Q$ and $P$ satisfy *equalized generative treatment* w.r.t. $\mathcal{D}_f$.

This definition captures the idea that the divergence experienced by each sensitive group should be approximately equal. In contrast to MGO or EGO, which constrain only proportions, EGT enforces a parity w.r.t. the actual metric that is being used to train or evaluate our generative model.

**Conditional closure.** Conceptually, the best way to approximate a target distribution $P$ while respecting $\delta$-EGT would be to treat each sensitive group independently. More specifically, for every $a \in \mathcal{A}$, one would learn a conditional distribution $Q_a$ that minimizes $\mathcal{D}_f(P_a \| Q_a)$, and then recombine the conditional models using the true proportions

$(\pi_a^P)_{a \in \mathcal{A}}$ of the target distribution. This procedure would ensure that each sensitive group is treated equally and that the final mixture aligns with $P$ as closely as possible. In practice, however, generative models are rarely designed in such a conditional manner. Instead, they typically produce distributions $Q$ as a whole, without the ability to freely optimize and reassemble conditionals. As a result, the typical family $\mathcal{Q}$ of candidate models used in practice seldom captures what would be achievable if sensitive group-level flexibility were available. To reason about this gap, we introduce the *conditional closure* of $\mathcal{Q}$, an augmented set of distributions that allows explicit recombination of conditionals.

**Definition 4.2.** Let $\mathcal{Q} \subseteq \mathcal{P}_\lambda(\mathcal{X})$ be a family of candidate models. For each $a \in \mathcal{A}$, define

$$\mathcal{Q}_a = \left\{ R \in \mathcal{P}_\lambda(\mathcal{X}_a) \,\middle|\, \exists Q \in \mathcal{Q} \text{ such that } Q_a = R \right\},$$

the set of group-$a$ conditional distributions realized by some model in $\mathcal{Q}$. Let also

$$\Delta_\mathcal{Q} = \left\{ (\pi_a)_{a \in \mathcal{A}} \in \Delta(\mathcal{A}) \,\middle|\, \begin{array}{l} \exists Q \in \mathcal{Q} \text{ such that} \\ (\pi_a^Q)_{a \in \mathcal{A}} = (\pi_a)_{a \in \mathcal{A}} \end{array} \right\}.$$

The *conditional closure* of $\mathcal{Q}$, denoted $\overline{\mathcal{Q}}_\mathcal{A}$, is

$$\overline{\mathcal{Q}}_\mathcal{A} = \left\{ \sum_{a \in \mathcal{A}} \pi_a R_a \,\middle|\, \begin{array}{l} (\pi_a)_{a \in \mathcal{A}} \in \Delta_\mathcal{Q}, \\ R_a \in \mathcal{Q}_a \; \forall a \in \mathcal{A} \end{array} \right\}.$$

Intuitively, $\overline{\mathcal{Q}}_\mathcal{A}$ can be seen as the completed version of $\mathcal{Q}$ for the fairness-constrained problem. It extends $\mathcal{Q}$ by allowing to select conditional models for each sensitive group from $\mathcal{Q}$ and then reassemble them under any group proportions realizable by $\mathcal{Q}$. In this sense, $\overline{\mathcal{Q}}_\mathcal{A}$ represents a best-case modeling scenario. If the original set $\mathcal{Q}$ has been explicitly designed to support attribute-conditional subdivision, then we may have $\mathcal{Q} = \overline{\mathcal{Q}}_\mathcal{A}$. In general, however, $\overline{\mathcal{Q}}_\mathcal{A}$ strictly contains $\mathcal{Q}$, since most generative models are not conditionally structured for sensitive attributes. The purpose of this closure is not to define an implementable algorithm, but to provide a technical baseline for the lower-bound argument. Theorem 4.3 shows that, under $\delta$-EGT, the global divergence $\mathcal{D}_f(P \| Q)$ cannot be made smaller than the largest group-conditional divergence achieved by the best model in the conditional closure, up to the tolerance $\delta$.

**Theorem 4.3.** *Let $P \in \mathcal{P}_\lambda(\mathcal{X})$ be a non-trivial target probability distribution and $f$ be a function such that $\mathcal{D}_f$ defines an $f$-divergence. Let $\mathcal{Q}$ be set of candidate probability distributions satisfying MGO with $P$ and such that there exists $Q^\star \in \arg\min_{Q \in \overline{\mathcal{Q}}_\mathcal{A}} \mathcal{D}_f(P \| Q)$. Then for any $\delta > 0$, if $Q \in \mathcal{Q}$ and $P$ satisfy $\delta$-EGT w.r.t. $\mathcal{D}_f$, then*

$$\mathcal{D}_f(P \| Q) \geq \max_{a \in \mathcal{A}} \mathcal{D}_f(P_a \| Q_a^\star) - \delta. \tag{1}$$

This result should be interpreted as a lower-bound statement rather than as a tight characterization. It shows that, under EGT, any candidate model must pay attention to the worst group-conditional divergence: if this divergence is large in the best conditionally recombined model, then no $\delta$-EGT model in $\mathcal{Q}$ can have arbitrarily small global divergence. When this lower bound is close to tight, minimizing the worst conditional divergence becomes the natural objective. This motivates the Min–Max schemes studied next.

### 4.2. Min–Max as Candidate for Enforcing EGT

Before diving into empirical considerations in the next section, we first analyze the feasibility of training a model that satisfies EGT from a theoretical viewpoint. To do so, we consider an arbitrary target distribution $P$ and a set of candidate models $\mathcal{Q}$. We would like to find the distributions $Q \in \mathcal{Q}$ that approximate $P$ the best, but also need to enforce $\delta$-EGT. In this context, one direct strategy would be to solve the following regularized problem

$$\min_{Q \in \mathcal{Q}} \mathcal{D}_f(P\|Q) + \lambda \sum_{a,a' \in \mathcal{A}} \left| \mathcal{D}_f(P_a\|Q_a) - \mathcal{D}_f(P_{a'}\|Q_{a'}) \right|, \tag{2}$$

where $\lambda \geq 0$ controls the strength of the EGT regularization. $\lambda = 0$ recovers standard divergence minimization, while larger values place increasing emphasis on balancing the conditional divergences across groups. However, while conceptually appealing, directly enforcing EGT via $f$-divergence regularization is impractical in modern training loops. $f$-divergences are inherently one-sided, yielding either a tractable minimization objective (variational) or a maximization surrogate (adversarial), but not both simultaneously (Huszár, 2015; Arjovsky & Bottou, 2017).

To circumvent the limitations of directly enforcing EGT through regularization, the next natural candidate is to rewrite the problem as a Min–Max optimization scheme,

$$\min_{Q \in \mathcal{Q}} \max_{a \in \mathcal{A}} \mathcal{D}_f\left(P_a\|Q_a\right). \tag{3}$$

Instead of minimizing a regularized $f$-divergence, this objective minimizes the worst conditional divergence across groups, pushing the model toward balancing its errors. In view of Theorem 4.3, this objective targets the quantity that controls the lower bound, while avoiding the stronger claim that EGT is exactly equivalent to Min–Max optimization in all model classes. Unlike direct EGT regularization, Min–Max schemes are practical and already appear in several distributionally robust optimization and group-robust learning settings (Oren et al., 2019; Sagawa* et al., 2019). In the next section, we therefore study this kind of scheme as a practical approximation to EGT rather than relying on direct regularization.

## 5. Improved Fairness Through EGT

This section evaluates whether optimization strategies that balance training losses across sensitive groups also improve fairness under our EGT criterion on evaluation metrics. For any group-wise metric $M$ (e.g., precision, recall, or FID), we report its EGT disparity as

$$\delta\text{-}M = \max_{a,a' \in \mathcal{A}} \left| M_a - M_{a'} \right|. \tag{4}$$

Thus, $P$, $R$ and $PR$ in the tables denote precision, recall and their sum, respectively. While precision and recall themselves should be maximized, $\delta$-P, $\delta$-R, and $\delta$-PR are disparities across sensitive groups and should be minimized.

**Compared methods.** Three strategies are considered. *Min–Max training* is the only approach that explicitly targets EGT by minimizing the worst group-wise conditional divergence (Theorem 4.3). This objective is popular in settings where equalizing (or protecting) worst-group *training* losses is observed to improve robustness. In generative modeling, however, such equalization is typically reported at the level of the optimized loss, and its impact on sample-based evaluation fairness metrics is less understood. *Conditional training* provides direct control of MGO/EGO at generation time by learning attribute-conditional distributions. While it does not optimize EGT explicitly, it mitigates the brittleness highlighted by Theorem 3.1 by preventing the training objective from collapsing onto a single group. In this sense, conditioning can be seen as a practical mechanism to avoid pathological error concentration, even though it does not enforce EGT. Finally, *reweighted loss* is a widely used baseline in fair generative modeling: it rescales per-sample losses to emphasize underrepresented groups and is primarily designed to improve the proportion-based criteria EGO. Importantly, RW comes with no guarantee for EGT and can even *increase* conditional quality gaps. This comparison therefore separates methods that control proportions (reweighting and conditional generation, which primarily address EGO/MGO) from the method that explicitly controls worst-group generation quality (Min–Max, which targets EGT).

**Methods and implementation details.** Since the model families and datasets are already introduced in Section 3 and fully detailed in Appendix B, the focus here is on how the three training strategies are implemented. For diffusion, the baseline starts from the public EDM checkpoints (VP/VE) of Karras et al. (2022). For text, LoRA adapters are fine-tuned on biography generation for LLaMA-3.2-Chat (1B/3B) and Gemma-3 (4B, pretrained and instruction-tuned). Conditional variants are obtained by adding standard class-conditioning to diffusion and by training LLMs under an attribute-constrained prompt (e.g., "a male/female person"). Reweighting rescales the negative log-likelihood (LLMs) or denoising/reconstruction objective (diffusion)

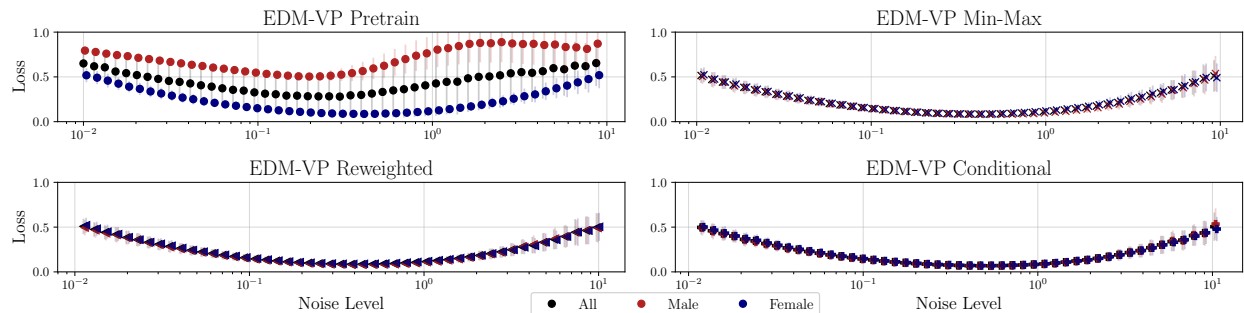

*Figure 3.* Estimated denoising losses per noise level for EDM-VP trained on FFHQ. Baseline exhibits a persistent gap between male and female groups. Reweighting and Min–Max reduce the gap, while conditional training almost eliminates it.

with group-dependent weights equal to $\pi_a^P/\pi_a^Q$. Although EDM is not the newest image-generation architecture, it remains a relevant and controlled benchmark for our goal: strong public VP/VE checkpoints are available on FFHQ, the training objective is explicit and stable, and the setting lets us compare optimization strategies without confounding architectural changes.

For Min–Max, we do not estimate an $f$-divergence explicitly at every update. Instead, we use the training loss optimized by each generative model as a stochastic proxy for the corresponding divergence: the denoising MSE for diffusion models and the next-token negative log-likelihood for LLMs. At each iteration, we compute empirical group losses $\ell_a$ on the current minibatch, maintain smoothed losses $L_a \leftarrow \alpha L_a + (1-\alpha)\ell_a$, select $a^\star = \arg\max_{a\in\mathcal{A}} L_a$, and update the model with the gradient of the worst-group loss $\nabla_\theta \ell_{a^\star}$. Full pseudocode, including the per-noise-level variant used for diffusion, is provided in Algorithms 1 and 2.

Operationally, this means that Min–Max repeatedly identifies the group currently associated with the largest smoothed loss and allocates the update to this group. It is therefore a direct way to prioritize the generations that are currently of lowest quality under the optimized surrogate.

**Min–Max training and stability improvements.** For diffusion, Min–Max must account for the multi-noise nature of training: rather than selecting a single worst group globally, the max is taken *per noise level* (Algorithm 1), ensuring that no subset of the diffusion trajectory disproportionately harms a group. This "per-noise" maximization is essential in practice because different groups can dominate the loss at different noise regimes. For both diffusion and LLM fine-tuning, the worst-group selection is stabilized via an exponential moving average (EMA) of group-wise losses (Algorithm 2). This EMA acts as a short memory that reduces batch-to-batch variance, prevents rapid oscillations of the maximizer, and avoids degenerate updates driven by a single noisy minibatch. Empirically, it yields markedly more stable optimization than naive per-batch maximization, especially in the low-batch regime of LLM fine-tuning.

**Image generation: closing loss gaps is not enough.** For EDM-VP on FFHQ, Figure 3 reports estimated denoising losses across noise levels. The pretrained baseline exhibits persistent loss gaps between male and female groups, and all three strategies substantially reduce these gaps (average losses are reported in Appendix Table 6). This observation matches prior evidence in robust learning: Min–Max objectives can effectively *equalize the optimized training loss* across groups. However, balancing the training loss does not guarantee balanced performance on evaluation metrics. The training objective is only a surrogate for generative quality, and improvements in loss can translate imperfectly to sample-based evaluation metrics. Therefore, closing subgroup loss gaps should be viewed as a useful diagnostic, but not a sufficient condition for improved EGT at evaluation.

**Image generation: improvements in EGT metrics.** Table 1 summarizes EGT disparities on precision, recall, and their sum $P+R$ (a valid $f$-divergence by linearity), together with $\delta$-FID for completeness. On EDM-VP, Min–Max yields a clear and stable gain: $\delta$-PR drops from 2.2 (pretrained) to 0.3, with $\delta$-P and $\delta$-R collapsing from 2.0/4.2 to 0.1/0.2. In contrast, conditional training can reduce $\delta$-P but may worsen $\delta$-R (and therefore $\delta$-PR), and reweighting sometimes improves disparities but not consistently across settings. Overall, explicitly optimizing the worst-group divergence appears necessary to obtain systematic improvements under EGT.

**Generality of the results.** We report FFHQ in the main text because it provides a direct image-generation counterpart to the Wikipedia biography setting: both tasks involve generation quality across gender-defined groups, allowing us to compare the same fairness question across modalities. To verify that the observed behavior is not specific to FFHQ, Appendix B.3.4 extends the diffusion study to additional image datasets with different label cardinalities. In particular, we evaluate the same EDM-VP/VE models and training strategies on CIFAR-10 and AFHQv2, and compare them side by side with FFHQ.

**Text generation: larger initial gaps and partial improve-**

*Table 1.* Evaluation of EDM models on FFHQ. We report subgroup disparities in precision ($\delta$-P), recall ($\delta$-R), their sum ($\delta$-PR), and FID ($\delta$-FID), all in percentage points. For each metric and generation setting, best values are bold and values improving on the unconstrained pretrained baseline are underlined.

| Model | Method | $\delta$-P | $\delta$-R | $\delta$-PR | $\delta$-FID |
|---|---|---|---|---|---|
| VP | Pretrained | 2.0 | 4.2 | 2.2 | 0.34 |
| | Conditional | **1.6** | 6.8 | 5.2 | **_0.08_** |
| | Reweighted | **1.8** | **2.3** | **0.6** | 0.44 |
| | Min–Max | **_0.1_** | **_0.2_** | **_0.3_** | 0.24 |
| VE | Pretrained | 2.1 | 4.4 | 2.3 | 0.35 |
| | Conditional | 0.9 | 2.7 | 3.5 | **0.11** |
| | Reweighted | 0.8 | 3.0 | 2.2 | **_0.04_** |
| | Min–Max | **_0.3_** | **_1.1_** | **_0.8_** | 0.33 |

*Table 2.* Evaluation of LLaMA-3.2-Chat models fine-tuned on Wikipedia Biographies. We report subgroup disparities in precision ($\delta$-P), recall ($\delta$-R), and their sum ($\delta$-PR), all in percentage points. For each metric and generation setting, best values are bold and values improving on the unconstrained pretrained baseline are underlined.

| Model | Method | $\delta$-P | $\delta$-R | $\delta$-PR |
|---|---|---|---|---|
| LLaMA-3.2-Chat 1B | Pretrained | 21.18 | 11.46 | 32.65 |
| | Conditional | **_19.09_** | **7.81** | **_26.90_** |
| | Reweighted | 21.29 | 9.22 | _30.50_ |
| | Min–Max | _19.55_ | **_7.73_** | _27.28_ |
| LLaMA-3.2-Chat 3B | Pretrained | 25.37 | 15.06 | 40.43 |
| | Conditional | **24.31** | **_12.26_** | **36.57** |
| | Reweighted | _22.88_ | _12.29_ | _35.16_ |
| | Min–Max | **_22.27_** | 12.87 | **_35.13_** |
| Gemma-3 4B (pt) | Pretrained | 20.90 | 12.13 | 33.03 |
| | Conditional | 23.17 | **_11.08_** | 34.25 |
| | Reweighted | _19.67_ | _11.71_ | _31.38_ |
| | Min–Max | **_19.13_** | _11.67_ | **_30.79_** |
| Gemma-3 4B (it) | Pretrained | 22.30 | 6.50 | 28.81 |
| | Conditional | 24.16 | 8.41 | 32.57 |
| | Reweighted | **_20.91_** | **_6.20_** | **_27.11_** |
| | Min–Max | 21.65 | 6.28 | _27.93_ |

**ments.** Results on Wikipedia Biographies (Table 2) exhibit substantially larger initial disparities than diffusion models. As in the image setting, conditional training and reweighting yield mixed outcomes: improvements are sometimes observed, but they are not systematic. Min–Max again provides the most consistent reduction in $\delta$-P, $\delta$-R, and $\delta$-PR, although the magnitude of improvement is smaller in absolute terms. A key practical difference is computational: LLM fine-tuning is significantly more expensive than diffusion fine-tuning in our setup, which limits the extent to which optimization can reshape representations.

**Takeaway.** Overall, balancing group-wise training losses is not sufficient to guarantee EGT improvements on evaluation divergences. Among the tested strategies, Min–Max is the only method that *systematically* reduces EGT dis-

parities across modalities, making it a promising approach for fairness beyond proportion matching. In addition, the empirical correlations in Table 3 computed across all experiments confirm the brittleness result from Section 3: $\delta$-MGO and $\delta$-EGO show only negligible association with $\delta$-P, $\delta$-R, and $\delta$-PR, so matching proportions alone provides little predictive control over group-wise conditional quality.

*Table 3.* Correlations between different $\delta$-MGO, $\delta$-EGO, $\delta$-Precision ($\delta$-P), $\delta$-Recall ($\delta$-R), $\delta$-Precision/Recall ($\delta$-PR), and $\delta$-FID, computed over all diffusion and LLM experiments.

| | $\delta$-MGO | $\delta$-EGO | $\delta$-P | $\delta$-R | $\delta$-PR | $\delta$-FID |
|---|---|---|---|---|---|---|
| $\delta$-MGO | 1.00 | -1.00 | 0.04 | -0.10 | -0.04 | -0.20 |
| $\delta$-EGO | -1.00 | 1.00 | -0.18 | -0.08 | -0.14 | 0.20 |

*Disclaimer.* These fairness improvements do not come for free. As discussed in Section 4, reducing disparities typically comes at the expense of the best-performing group. This trade-off is visible in Appendix B. For instance, under MGO with EDM-VP, Min–Max reduces $\delta$-PR from 2.2 to 0.3, but recall for the strongest subgroup drops from 91.7 to 86.6. Similarly, conditional training achieves strong $\delta$-FID (down to 0.08) but can substantially increase the overall FID (from 2.34 to 2.75). Overall, EGT gains should therefore be interpreted as a redistribution of errors across groups rather than a uniform improvement for every subgroup.

## 6. Concluding Remarks

In this work, we demonstrated that existing proportion-based criteria for fairness in generative modeling are inherently brittle: even when perfectly satisfied, a model's output quality can remain arbitrarily unbalanced across sensitive groups. To address this, we introduced equalized generative treatment (EGT), a fairness definition that enforces comparable $f$-divergences across groups, and studied its applicability. While $f$-divergences are among the most widely used metrics in generative models, alternatives such as the FID or integral probability metrics also exist and merit further investigation. In particular, it would be interesting to examine whether our analysis (especially results analogous to Theorems 3.1 and 4.3) remains valid in these alternative settings. Similarly, we expect that our results could extend to training schemes that directly minimize Wasserstein distances, such as Wasserstein GANs. We leave these explorations to future work.

## Acknowledgments

Rafael is partially supported by the French National Research Agency and the French Ministry of Research and Higher Education and by the CNRS through the International Emerging Action (IEA) program. This work was

also supported by the French government under management of Agence Nationale de la Recherche as part of the "Investissements d'avenir" program, PR[AI]RIE reference ANR-19-P3IA-0001, PR[AI]RIE-PSAI reference 23-IACL-0008, and by ANR JCJC TuLIP, reference ANR-25-CE23-1663. This work was granted access to the HPC resources of IDRIS under the allocations 2025-AD011014053R2 and 2025-A0181016159 made by GENCI.

## Impact Statement

This work introduces a definition of fairness tailored to generative models that explicitly enforces comparable generation quality across sensitive groups. By going beyond existing, less restrictive formulations, it provides a more principled framework for identifying and quantifying latent disparities in model behavior, with the goal of promoting more equitable systems. At the same time, we emphasize the inherent trade-offs associated with imposing such fairness constraints. In particular, improving parity across groups may lead to reductions in aggregate performance and a redistribution of errors toward groups that were previously better served. These effects underscore the need for careful, context-aware evaluation when deploying fairness-aware generative models in practice.

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

# A. Mathematical Supplementary Material

## A.1. Useful Lemma on the Surjectivity of $\mathcal{D}_f$

We begin with a statement that is of independent interest, as it establishes the surjectivity of the mapping $\mathcal{D}_f(R, \cdot)$ under suitable continuity assumptions on $f$. This result is stated in Lemma A.1.

**Lemma A.1.** *Let $f : [0, +\infty) \to (-\infty, +\infty]$ be a continuous function such that $\mathcal{D}_f$ defines an $f$-divergence. Then for any $R \in \mathcal{P}_\lambda(\mathcal{X})$, the map $\mathcal{D}_f(R, \cdot) : \{Q \in \mathcal{P}_\lambda(\mathcal{X}) \mid \text{SUPP}(R) = \text{SUPP}(Q)\} \to \mathbb{R}_+$ is surjective onto the interval $(0, f(0) + \bar{f}(+\infty))$.*

*Proof.* Let $R \in \mathcal{P}_\lambda(\mathcal{X})$, and let $f : [0, +\infty) \to (-\infty, +\infty]$ be a convex and continuous function with $f(1) = 0$. Let us also fix $\alpha \in (0, 1)$ and $\beta \in (1, +\infty)$. Since $R$ is absolutely continuous with respect to the Lebesgue measure $\lambda$, its cumulative density function is also continuous on $\mathbb{R}$. Accordingly, by definition of the cumulative density function and by the intermediate value theorem, there exists $A_\alpha \in \mathcal{B}(\mathcal{X})$ such that $R(A_\alpha) = \alpha$.

*1) Construction of an auxiliary mapping $\phi$.* Let us denote by $r := \frac{dR}{d\lambda}$ the probability density function of $R$ with respect to $\lambda$. Thanks to the above, we can define $Q_\alpha^\beta$ the probability distribution in $\mathcal{P}_\lambda(\mathcal{X})$ that admits a probability density function $\frac{dQ_\alpha^\beta}{d\lambda} = q_\alpha^\beta$ defined for all $x \in \mathcal{X}$ as

$$q_\alpha^\beta(x) := \frac{1}{\beta}\left(\frac{1}{\alpha}r(x)\mathbb{1}_{A_\alpha}(x)\right) + \left(1 - \frac{1}{\beta}\right)\left(\frac{1}{1-\alpha}r(x)\mathbb{1}_{\mathcal{X}\setminus A_\alpha}(x)\right). \tag{5}$$

By construction, since $\beta > 1$ and $\alpha \in (0, 1)$, $q_\alpha^\beta$ is a valid probability density function and $Q_\alpha^\beta \in \mathcal{P}_\lambda(\mathcal{X})$. Furthermore, also by construction we have $\text{SUPP}(R) = \text{SUPP}(Q_\alpha^\beta)$ (see Appendix A.1.1 for more details). Hence, using the alternative characterization of $f$-divergences in the special case of matching support (see e.g. (Polyanskiy & Wu, 2025, Chapter 7)), we have

$$\mathcal{D}_f(R\|Q_\alpha^\beta) = \int_\mathcal{X} f\left(\frac{r(x)}{q_\alpha^\beta(x)}\right) q_\alpha^\beta(x)\, d\lambda(x) \quad \text{with the convention } f(\tfrac{0}{0}) \times 0 = 0.$$

Using the above, and by definition of $Q_\alpha^\beta$, we thus have

$$\mathcal{D}_f(R\|Q_\alpha^\beta) = \int_{A_\alpha} f\left(\frac{r(x)}{\frac{r(x)}{\beta\alpha}}\right)\frac{r(x)}{\alpha\beta}\, d\lambda(x) + \int_{\mathcal{X}\setminus A_\alpha} f\left(\frac{r(x)}{r(x)\frac{\beta-1}{(1-\alpha)\beta}}\right)\frac{\beta-1}{\beta(1-\alpha)}r(x)\, d\lambda(x)$$

$$= \int_{A_\alpha} \frac{f(\beta\alpha)}{\alpha\beta}r(x)\, d\lambda(x) + \int_{\mathcal{X}\setminus A_\alpha} f\left(\frac{(1-\alpha)\beta}{\beta-1}\right)\frac{\beta-1}{\beta(1-\alpha)}r(x)\, d\lambda(x).$$

Furthermore, by linearity of the integral and by construction of $A_\alpha$, we have

$$\mathcal{D}_f(R\|Q_\alpha^\beta) = \frac{1}{\alpha\beta}f(\beta\alpha)\int_{A_\alpha} r(x)\, d\lambda(x) + \frac{\beta-1}{\beta(1-\alpha)}f\left(\frac{(1-\alpha)\beta}{\beta-1}\right)\int_{\mathcal{X}\setminus A_\alpha} r(x)\, d\lambda(x)$$

$$= \frac{1}{\alpha\beta}f(\beta\alpha)R(A_\alpha) + \frac{\beta-1}{\beta(1-\alpha)}f\left(\frac{(1-\alpha)\beta}{\beta-1}\right)R(\mathcal{X}\setminus A_\alpha)$$

$$= \frac{1}{\beta}f(\beta\alpha) + \frac{\beta-1}{\beta}f\left(\frac{(1-\alpha)\beta}{\beta-1}\right). \tag{6}$$

Since the $\alpha$, and $\beta$ have been chosen arbitrarily. The above construction holds for any $(\alpha, \beta) \in (0, 1) \times (1, +\infty)$. Thus, we can define $\phi : (0, 1) \times (1, +\infty) \to \mathbb{R}_+$ the mapping such that

$$\phi(\alpha, \beta) := \mathcal{D}_f(R\|Q_\alpha^\beta) = \frac{1}{\beta}f(\beta\alpha) + \frac{\beta-1}{\beta}f\left(\frac{(1-\alpha)\beta}{\beta-1}\right), \quad \forall(\alpha, \beta) \in (0, 1) \times (1, +\infty).$$

As $\{Q_\alpha^\beta \mid (\alpha, \beta) \in (0,1) \times (1, +\infty)\} \subseteq \{Q \in \mathcal{P}_\lambda(\mathcal{X}) \mid \text{SUPP}(R) = \text{SUPP}(Q)\}$, to get the expected result it suffices to show that $\phi$ is surjective onto the interval $(0, f(0) + \bar{f}(+\infty))$.

*2) Studying the surjectivity of $\phi$.* As $f$ is a continuous function, by composition of continuous functions, $\phi$ is jointly continuous on $(0,1) \times (1, +\infty)$. Accordingly, still using the intermediate value theorem, $\phi$ is surjective onto $[\phi(\alpha_a, \beta_a), \phi(\alpha_b, \beta_b)]$ for any $\alpha_a, \alpha_b \in (0,1)$ and $\beta_a, \beta_b \in (1, +\infty)$. In particular, since this holds for any choice of $\alpha_a, \alpha_b, \beta_a$, and $\beta_b$ we also have that $\phi$ is surjective onto $[\phi(1/2, 2), \lim_{\substack{\alpha \to 1 \\ \beta \to +\infty}} \phi(\alpha, \beta))$. To conclude, we just need to compute each of these terms:

- $\phi(1/2, 2) = \frac{1}{2}f(1) + \frac{1}{2}f(1) = 0$, and

- $\lim_{\substack{\alpha \to 1 \\ \beta \to +\infty}} \phi(\alpha, \beta) = \lim_{\beta \to \infty} \frac{1}{\beta}f(\beta) + \lim_{\beta \to +\infty} \frac{\beta-1}{\beta}f(0) = \bar{f}(+\infty) + f(0).$

*3) Conclusion.* By surjectivity of $\phi$ and by construction of $\{Q_\alpha^\beta \mid (\alpha, \beta) \in (0,1) \times (1, +\infty)\}$, we just showed that for any $y \in (0, f(0) + \bar{f}(\infty))$, there exists $Q \in \{Q \in \mathcal{P}_\lambda(\mathcal{X}) \mid \text{SUPP}(R) = \text{SUPP}(Q)\}$ such that $\mathcal{D}_f(R, Q) = y$, which concludes the proof. $\qquad\square$

### A.1.1. ADDITIONAL SANITY CHECKS FOR THE CONSTRUCTION OF $Q_\alpha^\beta$ IN LEMMA A.1

**Well-Definiteness of the Distribution.** Let us first show that for any $(\alpha, \beta) \in (0,1) \times (1, +\infty)$, $q_\alpha^\beta$ is a valid probability density function. For this, first note that the terms $1/\beta$, $1 - 1/\beta$, $1/\alpha$ and $1/(1-\alpha)$ are all positive. Furthermore, $r$ is itself a probability density function by definition, hence non-negative. Hence, by construction $q_\alpha^\beta$ is a non-negative function. Also note that, for any $(\alpha, \beta) \in (0,1) \times (1, +\infty)$, integrating against $\lambda$ over $\mathcal{X}$, we get

$$\int_{\mathcal{X}} q_\alpha^\beta(x)\, d\lambda(x) = \int_{\mathcal{X}} \frac{1}{\beta}\left(\frac{1}{\alpha}r(x)\mathbb{1}_{A_\alpha}(x)\right) + \left(1 - \frac{1}{\beta}\right)\left(\frac{1}{1-\alpha}r(x)\mathbb{1}_{\mathcal{X}\backslash A_\alpha}(x)\right)d\lambda(x)$$

$$= \int_{A_\alpha} \frac{1}{\beta\alpha}r(x)d\lambda(x) + \int_{\mathcal{X}\backslash A_\alpha}\left(1 - \frac{1}{\beta}\right)\frac{1}{1-\alpha}r(x)d\lambda(x)$$

Which by linearity of the integral and definition of $r$ and $A_\alpha$ gives

$$= \frac{1}{\beta\alpha}R(A_\alpha) + \left(1 - \frac{1}{\beta}\right)\frac{1}{1-\alpha}R(\mathcal{X}\backslash A_\alpha) = \frac{1}{\beta} + \left(1 - \frac{1}{\beta}\right) = 1.$$

**Matching Supports.** Let us now show that $\text{SUPP}(Q_\alpha^\beta) = \text{SUPP}(R)$. To do so let us first consider $x \notin \text{SUPP}(R)$, by definition of $q_\alpha^\beta$ we have

$$q_\alpha^\beta(x) = \frac{1}{\beta\alpha}r(x)\mathbb{1}_{A_\alpha}(x) + \left(1 - \frac{1}{\beta}\right)\frac{1}{1-\alpha}r(x)\mathbb{1}_{\mathcal{X}\backslash A_\alpha}(x).$$

Since $x \notin \text{SUPP}(R)$, we have $r(x) = 0$, hence $q_\alpha^\beta(x) = 0$. Accordingly, $x \notin \text{SUPP}(Q_\alpha^\beta)$. This means by contrapositive that $\text{SUPP}(Q_\alpha^\beta) \subset \text{SUPP}(R)$. Similarly, let $x \notin \text{SUPP}(Q_\alpha^\beta)$ we have

$$q_\alpha^\beta(x) = \frac{1}{\beta\alpha}r(x)\mathbb{1}_{A_\alpha}(x) + \left(1 - \frac{1}{\beta}\right)\frac{1}{1-\alpha}r(x)\mathbb{1}_{\mathcal{X}\backslash A_\alpha}(x) = 0.$$

Since all terms $1/\beta$, $1 - 1/\beta$, $1/\alpha$ and $1/(1-\alpha)$ are positive, this means that $r(x) = 0$. Hence $x \notin \text{SUPP}(R)$, which gives us $\text{SUPP}(R) \subset \text{SUPP}(Q_\alpha^\beta)$.

### A.2. Proof of Theorem 1

Before proceeding to the proof of Theorem 3.1, we state a central lemma that decomposes the $f$-divergence between two measures $P$ and $Q$ in terms of a linear combination of the $f$-divergence between their marginals. The result is given in Lemma A.2.

### A.2.1. Preliminary Lemma

**Lemma A.2.** *Let $P \in \mathcal{P}_\lambda(\mathcal{X})$ be a non-trivial target probability distribution, $Q \in \mathcal{P}_\lambda(\mathcal{X})$, and $f$ be function such that $\mathcal{D}_f$ defines an $f$-divergence. If $P$ and $Q$ have matching generative odds, then*

$$\mathcal{D}_f(P\|Q) = \sum_{a\in\mathcal{A}} \pi_a^Q \mathcal{D}_f(P_a\|Q_a),$$

*where the decomposition according to $\mathcal{A}$ is as defined in Section 2.2.*

*Proof.* Let $P \in \mathcal{P}_\lambda(\mathcal{X})$ be a target probability distribution and $f$ be a function such that $\mathcal{D}_f$ defines an $f$-divergence. Let $Q \in \mathcal{P}_\lambda(\mathcal{X})$, such that $P$ and $Q$ have matching generative odds. By definition of the attribute mapping $\psi$, $\{\mathcal{X}_a \mid a \in \mathcal{A}\}$ is a partition of $\mathcal{X}$. Hence, denoting $p = \frac{dP}{d\lambda}$ and $q = \frac{dQ}{d\lambda}$ the respective probability density functions of $P$ and $Q$, we have

$$\mathcal{D}_f(P\|Q) = \int_{\text{SUPP}(Q)} f\left(\frac{p(x)}{q(x)}\right) q(x)d\lambda(x) + \bar{f}(+\infty)P(\mathcal{X} \setminus \text{SUPP}(Q))$$

$$= \sum_{a\in\mathcal{A}} \int_{\mathcal{X}_a \cap \text{SUPP}(Q)} f\left(\frac{p(x)}{q(x)}\right) q(x)d\lambda(x) + \bar{f}(+\infty)P(\mathcal{X} \setminus \text{SUPP}(Q)).$$

Note that, for any $a \in \mathcal{A}$, by definition we have

$$\mathcal{X}_a \cap \text{SUPP}(Q) := \{x \in \mathcal{X}_a \mid q(x) > 0\} = \{x \in \mathcal{X} \mid q(x) \times \mathbb{1}_{\mathcal{X}_a}(x) > 0\}.$$

Now recall that we assumed $\pi_a^P > 0$ for any $a \in \mathcal{A}$. Furthermore, $Q$ is assumed to match the odds of $P$, meaning that $\pi_a^Q = \pi_a^P > 0$ for all $a \in \mathcal{A}$. Hence we have

$$\mathcal{X}_a \cap \text{SUPP}(Q) = \{x \in \mathcal{X} \mid q(x) \times \mathbb{1}_{\mathcal{X}_a}(x) > 0\}$$

$$= \{x \in \mathcal{X} \mid \tfrac{q(x)}{\pi_a^Q} \times \mathbb{1}_{\mathcal{X}_a}(x) := q_a(x) > 0\} = \text{SUPP}(Q_a). \tag{7}$$

Using (7) in the first decomposition of $\mathcal{D}_f$, we get

$$\mathcal{D}_f(P\|Q) = \sum_{a\in\mathcal{A}} \int_{\text{SUPP}(Q_a)} f\left(\frac{p(x)}{q(x)}\right) q(x)d\lambda(x) + \bar{f}(+\infty)P(\mathcal{X} \setminus \text{SUPP}(Q)).$$

Now recall that we can always rewrite $p$ and $q$ as mixtures of attribute-conditional probability density functions $p = \sum_{a\in\mathcal{A}} \pi_a^P p_a$ and $q = \sum_{a\in\mathcal{A}} \pi_a^Q q_a$ where for any $a \in \mathcal{A}$, $q_a(x) = p_a(x) = 0$ for all $x \notin \mathcal{X}_a$. Accordingly, the $f$-divergence between $P$ and $Q$ can be rewritten as

$$\mathcal{D}_f(P\|Q) = \sum_{a\in\mathcal{A}} \int_{\text{SUPP}(Q_a)} f\left(\frac{\pi_a^P p_a(x)}{\pi_a^Q q_a(x)}\right) \pi_a^Q q_a(x)d\lambda(x) + \bar{f}(+\infty)P(\mathcal{X} \setminus \text{SUPP}(Q)). \tag{8}$$

Note also that, by definition of the by definition of the attribute oracle, we have $P = \sum_{a\in\mathcal{A}} \pi_a^P P_a$ where $\text{SUPP}(P_a) \subset \mathcal{X}_a$. Using this we can rewrite $P(\mathcal{X} \setminus \text{SUPP}(Q))$ as

$$P(\mathcal{X} \setminus \text{SUPP}(Q)) = \sum_{a\in\mathcal{A}} \pi_a^P P_a(\mathcal{X} \setminus \text{SUPP}(Q)) = \sum_{a\in\mathcal{A}} \pi_a^P P_a(\mathcal{X}_a \setminus \text{SUPP}(Q))$$

$$= \sum_{a\in\mathcal{A}} \pi_a^P P_a(\mathcal{X}_a \setminus (\text{SUPP}(Q) \cap \mathcal{X}_a))$$

$$= \sum_{a\in\mathcal{A}} \pi_a^P P_a(\mathcal{X}_a \setminus \text{SUPP}(Q_a)).$$

Where the last line comes from using (7). Using the above in (8) we get

$$\mathcal{D}_f(P\|Q) = \sum_{a\in\mathcal{A}} \int_{\text{SUPP}(Q_a)} f\left(\frac{\pi_a^P p_a(x)}{\pi_a^Q q_a(x)}\right) \pi_a^Q q_a(x)d\lambda(x) + \bar{f}(+\infty) \sum_{a\in\mathcal{A}} \pi_a^P P_a(\mathcal{X}_a \setminus \text{SUPP}(Q_a)).$$

Finally, Using the matching odds property ,i.e., the fact that $\pi_a^P = \pi_a^Q$ for all $a \in \mathcal{A}$, we have

$$
\begin{aligned}
\mathcal{D}_f(P\|Q) &= \sum_{a \in \mathcal{A}} \int_{\text{Supp}(Q_a)} f\left(\frac{p_a(x)}{q_a(x)}\right) \pi_a^Q q_a(x) d\lambda(x) + \bar{f}(+\infty) \sum_{a \in \mathcal{A}} \pi_a^Q P_a\left(\mathcal{X}_a \setminus \text{Supp}(Q_a)\right) \\
&= \sum_{a \in \mathcal{A}} \pi_a^Q \left( \int_{\text{Supp}(Q_a)} f\left(\frac{p_a(x)}{q_a(x)}\right) q_a(x) d\lambda(x) + \bar{f}(+\infty) P_a\left(\mathcal{X}_a \setminus \text{Supp}(Q_a)\right) \right) \\
&= \sum_{a \in \mathcal{A}} \pi_a^Q \mathcal{D}_f(P_a\|Q_a), \text{ which concludes the proof.}
\end{aligned}
$$

$\square$

### A.2.2. PROOF OF THE THEOREM

We now turn to the proof of Theorem 3.1, restated below as Theorem A.3. The argument relies primarily on Lemma A.1 and Lemma A.2.

**Theorem A.3.** *Let $P \in \mathcal{P}_\lambda(\mathcal{X})$ be a target probability distribution satisfying equalized generative odds and $f$ be a continuous function such that $\mathcal{D}_f$ defines an $f$-divergence. For any $\epsilon \in (0, f(0) + \bar{f}(+\infty))$ and $\gamma \in (0, \epsilon)$, there exists $Q^\gamma \in \mathcal{S}_{\mathcal{D}_f}(P, \epsilon)$ satisfying matching generative odds with $P$, but for which there exists $\bar{a} \in \mathcal{A}$ such that*

$$
\mathcal{D}_f(P_{\bar{a}}\|Q_{\bar{a}}^\gamma) \geq \mathcal{D}_f(P_a\|Q_a^\gamma) + \gamma, \text{ for all } a \in \mathcal{A} \setminus \{\bar{a}\}.
$$

*Proof.* Let $P \in \mathcal{P}_\lambda(\mathcal{X})$ be a target probability distribution satisfying equalized generative odds and $f$ be a continuous function such that $\mathcal{D}_f$ defines an $f$-divergence. By definition of the attribute mapping $\psi$, $\{\mathcal{X}_a \mid a \in \mathcal{A}\}$ is a partition of $\mathcal{X}$. Hence we can rewrite $P = \sum_{a \in \mathcal{A}} \pi_a^P P_a$ with $\text{Supp}(P_a) \subset \mathcal{X}_a$. Let us now fix $\epsilon \in (0, f(0) + \bar{f}(+\infty))$, $\gamma \in (0, \epsilon)$ and let $\bar{a} \in \mathcal{A}$. Since $f$ is continuous, by Lemma A.1, we know there exists $Q_{\bar{a}}^\gamma \in \mathcal{P}_\lambda(\mathcal{X})$ such that $\text{Supp}(P_{\bar{a}}) = \text{Supp}(Q_{\bar{a}}^\gamma)$ and $\mathcal{D}_f(P_{\bar{a}}\|Q_{\bar{a}}^\gamma) = \epsilon + \gamma \frac{|\mathcal{A}|-1}{|\mathcal{A}|}$. Similarly, for any $a \in \mathcal{A} \setminus \{\bar{a}\}$ there exist a distribution $Q_a^\gamma \in \mathcal{P}_\lambda(\mathcal{X})$ such that $\text{Supp}(P_a) = \text{Supp}(Q_a^\gamma)$ and $\mathcal{D}_f(P_a\|Q_a^\gamma) = \epsilon - \frac{\gamma}{|\mathcal{A}|}$. Using these, we define $Q^\gamma$ as

$$
Q^\gamma = \frac{1}{|\mathcal{A}|} \sum_{a \in \mathcal{A}} Q_a^\gamma.
$$

Let us now consider $Q^\gamma$ as defined above. Recall that the target distribution $P$ is assumed to satisfy equalized odds, hence $\pi_a^P = \frac{1}{|\mathcal{A}|}$ for all $a \in \mathcal{A}$. Furthermore, by definition, $\pi_a^{Q^\gamma} = \frac{1}{|\mathcal{A}|} = \pi_a^P$ for all $a \in \mathcal{A}$, which means that $Q^\gamma$ satisfies matching odds with $P$. Hence, using Lemma A.2, we have

$$
\mathcal{D}_f(P\|Q^\gamma) = \sum_{a \in \mathcal{A}} \pi^{Q_a^\gamma} \mathcal{D}_f(P_a\|Q_a^\gamma) = \frac{1}{|\mathcal{A}|} \sum_{a \in \mathcal{A}} \mathcal{D}_f(P_a\|Q_a^\gamma) = \epsilon
$$

Nevertheless, by construction, we also have

$$
\mathcal{D}_f(P_{\bar{a}}\|Q_{\bar{a}}) \geq \mathcal{D}_f(P_a\|Q_a) + \gamma.
$$

Hence, by construction, we just designed a probability distribution $Q^\gamma$ satisfying the conditions of Theorem A.3. $\square$

### A.3. Proof of Theorem 2

We now present the proof of Theorem 4.3, restated below as Theorem A.4. The proof relies primarily on Lemma A.2 and the notion of conditional closure.

**Theorem A.4.** *Let $P \in \mathcal{P}_\lambda(\mathcal{X})$ be a target probability distribution and $f$ be a function such that $\mathcal{D}_f$ defines an $f$-divergence. Let $\mathcal{Q}$ be set of candidate probability distributions satisfying matching odds with respect to $P$ and such that there exists $Q^\star \in \arg\min_{Q \in \overline{\mathcal{Q}}_\mathcal{A}} \mathcal{D}_f(P\|Q)$. Then for any $\delta > 0$, if $Q \in \mathcal{Q}$ satisfies $\delta$-equalized generative treatment of $P$ with respect to $\mathcal{D}_f$, then*

$$
\mathcal{D}_f(P\|Q) \geq \max_{a \in \mathcal{A}} \mathcal{D}_f(P_a\|Q_a^\star) - \delta.
$$

*Proof.* Let $P \in \mathcal{P}_\lambda(\mathcal{X})$ be a target probability distribution satisfying equalized generative odds and $f$ be a function such that $\mathcal{D}_f$ defines an $f$-divergence. By definition of the attribute mapping $\psi$, $\{\mathcal{X}_a \mid a \in \mathcal{A}\}$ is a partition of $\mathcal{X}$. Hence we can rewrite $P = \sum_{a \in \mathcal{A}} \pi_a^P P_a$ with $\mathrm{SUPP}(P_a) \subset \mathcal{X}_a$. Let us now consider $Q \in \mathcal{Q}$. As all probability distributions in $\mathcal{Q}$ satisfy matching odds with respect to $P$, we use Lemma A.2 to write

$$\mathcal{D}_f(P\|Q) = \sum_{a \in \mathcal{A}} \pi_a^Q \mathcal{D}_f(P_a, Q_a). \tag{9}$$

*Reasoning ab absurdum.* Based on the above, we reason ab absurdum to obtain the expected result. Let us take $a_\# \in \arg\max_{a \in \mathcal{A}} \mathcal{D}_f(P_a\|Q_a^\star)$, where $Q^\star \in \arg\min_{Q \in \overline{\mathcal{Q}}_\mathcal{A}} \mathcal{D}_f(P\|Q)$, and assume that

$$\mathcal{D}_f(P\|Q) < \mathcal{D}_f\left(P_{a_\#}\|Q_{a_\#}^\star\right) - \delta. \tag{10}$$

Note that, by definition of $\delta$-equalized generative treatment, we know that

$$\mathcal{D}_f\left(P_{a_\#}\|Q_{a_\#}\right) - \mathcal{D}_f\left(P_a\|Q_a\right) \leq \delta, \ \forall a \in \mathcal{A}.$$

Hence, we also have $\mathcal{D}_f\left(P_{a_\#}\|Q_{a_\#}\right) \leq \mathcal{D}_f\left(P_a\|Q_a\right) + \delta, \ \forall a \in \mathcal{A}$. Multiplying both sides by $\pi_a^Q$ for all $a \in \mathcal{A}$ and summing the terms one gets

$$\sum_{a \in \mathcal{A}} \pi_a^Q \mathcal{D}_f\left(P_{a_\#}\|Q_{a_\#}\right) \leq \sum_{a \in \mathcal{A}} \pi_a^Q \left(\mathcal{D}_f\left(P_a\|Q_a\right) + \delta\right).$$

Finally, using the fact that $\sum_{a \in \mathcal{A}} \pi_a^Q = 1$ and the decomposition in (9), we obtain

$$\mathcal{D}_f\left(P_{a_\#}\|Q_{a_\#}\right) \leq \mathcal{D}_f\left(P\|Q\right) + \delta. \tag{11}$$

Now, let us consider the distribution $\tilde{Q} \in \mathcal{P}_\lambda(\mathcal{X})$ defined as

$$\tilde{Q} = \sum_{\substack{a \in \mathcal{A} \\ a \neq a_\#}} \pi_a^{Q^\star} Q_a^\star + \pi_{a_\#}^{Q^\star} Q_{a_\#}.$$

By definition of $\overline{\mathcal{Q}}^\mathcal{A}$, we have that $\tilde{Q} \in \overline{\mathcal{Q}}^\mathcal{A}$ and that $Q^\star$ satisfies matching generative odds with respect to $P$. Hence, $\tilde{Q}$ also satisfies matching generative odds with respect to $P$, which gives us (using Lemma A.2)

$$\mathcal{D}_f\left(P\|\tilde{Q}\right) = \sum_{\substack{a \in \mathcal{A} \\ a \neq a_\#}} \pi_a^{Q^\star} \mathcal{D}_f(P_a\|Q_a^\star) + \pi_{a_\#}^{Q^\star} \mathcal{D}_f(P_{a_\#}\|Q_{a_\#}).$$

Substituting successively (11) and (10) in the above, and using the fact that $Q^\star$ satisfies matching generative odds with respect to $P$ (to obtain the last equality), we have

$$\mathcal{D}_f\left(P\|\tilde{Q}\right) \leq \sum_{\substack{a \in \mathcal{A} \\ a \neq a_\#}} \pi_a^{Q^\star} \mathcal{D}_f(P_a\|Q_a^\star) + \pi_{a_\#}^{Q^\star} \left(\mathcal{D}_f(P\|Q) + \delta\right)$$

$$< \sum_{\substack{a \in \mathcal{A} \\ a \neq a_\#}} \pi_a^{Q^\star} \mathcal{D}_f(P_a\|Q_a^\star) + \pi_{a_\#}^{Q^\star} \mathcal{D}_f\left(P_{a_\#}\|Q_{a_\#}^\star\right)$$

$$< \sum_{a \in \mathcal{A}} \pi_a^{Q^\star} \mathcal{D}_f(P_a\|Q_a^\star) = \mathcal{D}_f\left(P\|Q^\star\right).$$

According to the above, we just constructed a probability distribution $\tilde{Q} \in \overline{\mathcal{Q}}^\mathcal{A}$ that has an $f$-divergence strictly smaller than $Q^\star$ with respect to $P$. By definition of $Q^\star$, this is impossible, hence contradicting our initial assumption that $\mathcal{D}_f(P\|Q) < \mathcal{D}_f\left(P_{a_\#}\|Q_{a_\#}^\star\right) - \delta$. In particular, this means that

$$\mathcal{D}_f(P\|Q) \geq \max_{a \in \mathcal{A}} \mathcal{D}_f(P_a\|Q_a^\star) - \delta.$$

$\square$

# B. Experimental Setup

## B.1. Building Illustrative Examples

In this section, we provide an illustrative examples. Two synthetic scenarios are constructed to highlight limitations of proportion-based fairness constraints when subgroup generation quality differs. Both examples are based on FFHQ (Karras et al., 2021), which provides face images with gender metadata. Samples are generated at resolution $64 \times 64$ using a pretrained unconditional EDM-VP model (Karras et al., 2022). To enforce MGO, the generated set is first filtered so that male/female proportions match the desired ratio. Disparities are then introduced *post hoc* in a controlled way.

**Quality gap (precision).** Random Gaussian noise is added to one group (or both) to create a difference in fidelity. In Example 1, mild noise is applied to both groups, leading to a moderate precision drop for both. In Example 2, strong noise is applied to male images while female images remain unchanged, producing a large precision gap.

**Diversity gap (recall).** To reduce diversity, a subset of generated images is duplicated and randomly horizontally flipped. In Example 1, $20\%$ of images are duplicated in both groups, yielding a mild recall drop for both. In Example 2, only male images are duplicated, creating a large recall gap.

Final global and subgroup precision/recall values are reported in Table 4. Although the two examples have similar global P&R, subgroup metrics differ substantially, illustrating that matching proportions (MGO/EGO) can hide large disparities in conditional generation quality.

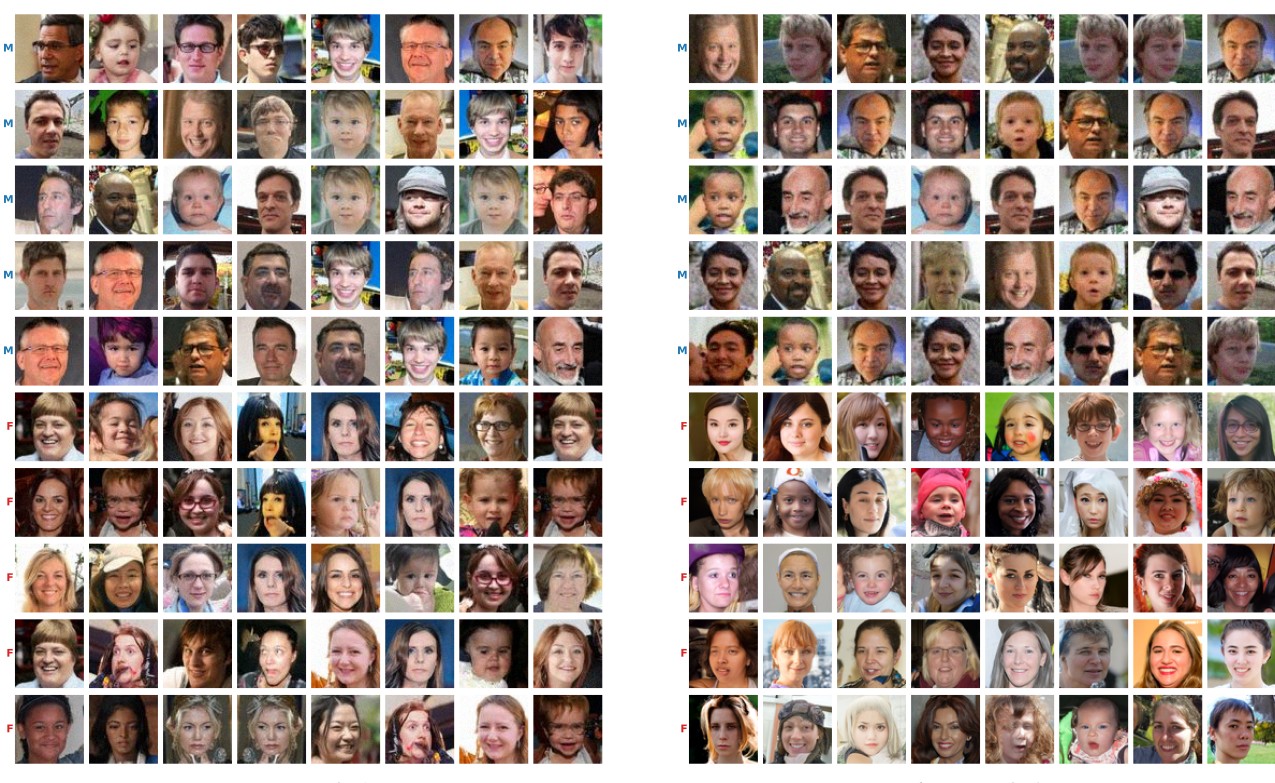

*(a)* Example 1          *(b)* Example 2

*Figure 4.* Samples from two FFHQ generators with similar global precision (79.06 in 4a and 77.12 in 4b) and similar global recall (60.02 in 4a and 60.31 in 4b). Despite similar global scores, subgroup precision and recall differ substantially. In Example 1, the model generates slightly noisy images for both classes. In Example 2, the model generates limited diversity and noisy images for the male class while generating high-quality and diverse samples for the female class.

*Table 4.* precision and recall for the two examples shown in Figure 4. The global precision and recall are similar, but the subgroup metrics differ significantly.

| Subset | Precision | Recall | M Recall | F Recall | M Precision | F Precision |
|---|---|---|---|---|---|---|
| Example 1 | 79.06 | 60.02 | 47.96 | 45.01 | 73.96 | 71.88 |
| Example 2 | 77.12 | 60.31 | 1.52 | 91.22 | 66.53 | 95.85 |

## B.2. Optimization Strategies and Baselines

In this section, we describe the three training strategies used throughout the experiments. They are architecture-agnostic: each can be instantiated for diffusion models by replacing $\mathcal{D}_f$ with the denoising loss proxy, and for LLMs by replacing it with the next-token negative log-likelihood. Direct EGT regularization is conceptually natural but impractical in modern training loops because it would require estimating and differentiating group-wise $f$-divergences at each update. We therefore compare three feasible alternatives: reweighted training, Min–Max training, and conditional training.

| Reweighted training | Min–Max training | Conditional training |
|---|---|---|
| $\min\limits_{Q \in \mathcal{Q}} \mathcal{D}_f\left(\sum\limits_{a \in \mathcal{A}} \frac{1}{\|\mathcal{A}\|} P_a \middle\| Q\right)$ | $\min\limits_{Q \in \mathcal{Q}} \max\limits_{a \in \mathcal{A}} \mathcal{D}_f(P_a \| Q_a)$ | $\min\limits_{\substack{(Q_a)_{a \in \mathcal{A}} \\ Q = \sum_a \pi_a^P Q_a \in \mathcal{Q}}} \mathcal{D}_f(P \| Q)$ |

**Reweighted training** rescales per-sample losses with group-dependent weights, following Choi et al. (2020). In our notation, samples from group $a$ are weighted proportionally to $\pi_a^P / \pi_a^Q$, where $\pi^Q$ denotes the empirical training proportion and $\pi^P$ the target proportion. This baseline is designed to reduce the dominance of frequent groups and often improves proportion-based criteria such as EGO, but it does not directly optimize EGT. For diffusion models, we apply the same idea independently across noise levels, following the practical spirit of Kim et al. (2023b).

**Min–Max training** is the only training strategy that directly targets the EGT objective. Instead of averaging losses across all samples, it selects the group with the largest smoothed loss and updates the model on this worst group. This corresponds to a stochastic implementation of the objective $\min_Q \max_a \mathcal{D}_f(P_a \| Q_a)$, where the optimized loss is used as a proxy for the group-conditional divergence.

**Conditional training** explicitly gives the sensitive attribute or class label to the model. At sampling time, this makes it possible to choose the group proportions manually, and therefore to enforce MGO or EGO by construction. Conditional training can also reduce conditional quality gaps by preventing the model from representing all groups through a single unconditional distribution. However, it does not directly minimize the worst group-wise divergence, and therefore does not guarantee EGT.

## B.3. Diffusion Models

### B.3.1. EXPERIMENTAL SET-UP

We use the public EDM codebase of Karras et al. (2022). EDM is a convenient benchmark for this study because it provides strong, reproducible VP and VE checkpoints on FFHQ, while keeping the architecture and training objective transparent

*Table 5.* Summary of the fairness controls provided by each baseline. Checkmarks indicate which criterion can be directly enforced by the method itself; EGT is checked only for Min–Max because it explicitly optimizes the worst group-wise quality proxy.

| Method | Stage | MGO | EGO | EGT |
|---|---|---|---|---|
| Standard unconditional | Training | ✗ | ✗ | ✗ |
| Rejection sampling (Zameshina et al., 2022) | Inference | ✓ | ✓ | ✗ |
| Conditional training | Training | ✓ | ✓ | ✗ |
| Reweighted training (Choi et al., 2020) | Training | ✗ | ✓ | ✗ |
| Min–Max training | Training | ✗ | ✗ | ✓ |
| Min–Max + rejection sampling | Both | ✓ | ✓ | ✓ |

enough to modify the loss at the group level. The two variants share the same U-Net backbone but differ in the noise parameterization and sampling schedule. Unless stated otherwise, the pretrained EDM-VP and EDM-VE checkpoints are used as initialization, and all fairness-oriented methods are implemented as fine-tuning procedures.

### B.3.2. TRAINING DIFFUSION MODELS

Diffusion training samples a clean image $x$, a noise level $\sigma$, and optimizes a weighted denoising objective. For the unconditional model, the denoiser $F_\theta(x, \sigma)$ is trained with

$$\mathcal{L}_{\text{Unconditional}}(\theta) = \mathbb{E}_{x \sim P, \sigma} \left[ w(\sigma) \left\| F_\theta(x, \sigma) - x \right\|^2 \right], \tag{12}$$

where $w(\sigma)$ denotes the EDM weighting at noise level $\sigma$. Conditional training uses the same denoising objective, but provides the group label $\psi(x)$ to the network:

$$\mathcal{L}_{\text{Conditional}}(\theta) = \mathbb{E}_{x \sim P, \sigma} \left[ w(\sigma) \left\| F_\theta(x, \psi(x), \sigma) - x \right\|^2 \right]. \tag{13}$$

The conditional model therefore shares almost all parameters with the unconditional one, with only a small additional embedding for the group label. This keeps the comparison focused on the training strategy rather than on model capacity.

For Min–Max training, applying a single maximization over groups is too coarse because the underperforming group can depend on the noise level. A group may dominate the loss at high noise while another dominates near the denoising end of the trajectory. We therefore select the worst group separately for each noise level and aggregate the corresponding losses before taking an optimizer step. This is the diffusion-specific version of the generic Min–Max procedure described in Algorithm 2, and it is detailed in Algorithm 1.

**Practical Min–Max Implementation.** The theoretical objective in Eq. equation 3 involves conditional $f$-divergences, but these quantities are not directly available inside modern stochastic training loops. In practice, we use the optimized training loss as a stochastic proxy for the corresponding divergence. For LLMs, this is the negative log-likelihood of the next-token prediction objective. For diffusion models, this is the denoising reconstruction loss at the sampled noise level. Given a minibatch $\mathcal{B}$ with labels in $\mathcal{A}$, we compute the empirical loss of each group,

$$\ell_a(\theta; \mathcal{B}) = \frac{1}{|\mathcal{B}_a|} \sum_{x \in \mathcal{B}_a} \ell(\theta; x), \qquad \mathcal{B}_a = \{x \in \mathcal{B} : \psi(x) = a\},$$

whenever $\mathcal{B}_a$ is non-empty. To avoid selecting a different maximizer because of minibatch noise, we maintain an EMA $L_a$ of these group losses and update only with the current loss of the group whose EMA is largest. For diffusion, the same idea is applied independently at each noise level, because the worst group can vary across the denoising trajectory.

---

**Algorithm 1** Min–Max diffusion training across noise levels

---

1: **Initialize** model parameters $\theta$ of $F_\theta$
2: **Initialize** moving averages $\{L_{a,\sigma}\}_{a\in\mathcal{A},\,\sigma\in\Sigma}$ (e.g., $L_{a,\sigma} \leftarrow 0$)
3: **for** each training iteration **do**
4:     Sample a minibatch $\mathcal{B}$ of data points $x$ with group labels $a \in \mathcal{A}$ and noise levels $\sigma \in \Sigma$
5:     Form $\mathcal{B}_{a,\sigma} = \{(x,\sigma') \in \mathcal{B} : \psi(x) = a,\ \sigma' = \sigma\}$
6:     Compute per-group, per-noise losses

$$\ell_{a,\sigma} = \frac{1}{|\mathcal{B}_{a,\sigma}|} \sum_{(x,\sigma)\in\mathcal{B}_{a,\sigma}} w(\sigma)\|F_\theta(x,\sigma) - x\|_2^2$$

    for non-empty $\mathcal{B}_{a,\sigma}$
7:     **for all** $(a,\sigma) \in \mathcal{A} \times \Sigma$ **do**
8:       **if** $\mathcal{B}_{a,\sigma}$ is non-empty **then**
9:         $L_{a,\sigma} \leftarrow \alpha L_{a,\sigma} + (1-\alpha)\,\ell_{a,\sigma}$
10:       **end if**
11:     **end for**
12:     **for all** $\sigma \in \Sigma$ **do**
13:       $a^\star(\sigma) \leftarrow \arg\max_{a\in\mathcal{A}} L_{a,\sigma}$
14:     **end for**
15:     $\mathcal{L} \leftarrow \sum_{\sigma\in\Sigma} \ell_{a^\star(\sigma),\,\sigma}$
16:     Compute gradient $g \leftarrow \nabla_\theta \mathcal{L}$
17:     Update parameters $\theta \leftarrow \text{OPTIMIZERSTEP}(\theta, g)$
18: **end for**

---

As training relies on mini-batch SGD, the number of samples from an underperforming sensitive group within a batch can be small and highly variable. This issue is further amplified when losses are tracked per noise level, which increases variance in the maximizer selection. To stabilize training and avoid noisy oscillations in the worst-group choice, an exponential moving average (EMA) is used to smooth group-wise losses over iterations. Algorithm 2 describes this EMA-based Min–Max strategy, which is applied both to diffusion training and to LLM fine-tuning.

---

**Algorithm 2** Min–Max training with EMA of group-wise losses

---

1: **Initialize** model parameters $\theta$ of $Q_\theta$
2: **Initialize** moving averages $\{L_a\}_{a\in\mathcal{A}}$ (e.g., $L_a \leftarrow 0$)
3: **for** each training iteration **do**
4:     Sample a minibatch $\mathcal{B}$ with group labels $a \in \mathcal{A}$
5:     Compute per-group losses $\{\ell_a\}_{a\in\mathcal{A}}$ on $\mathcal{B}$
6:     **for all** $a \in \mathcal{A}$ **do**
7:       $L_a \leftarrow \alpha L_a + (1-\alpha)\,\ell_a$
8:     **end for**
9:     $a^\star \leftarrow \arg\max_{a\in\mathcal{A}} L_a$
10:     Compute gradient $g \leftarrow \nabla_\theta \ell_{a^\star}$
11:     Update parameters $\theta \leftarrow \text{OPTIMIZERSTEP}(\theta, g)$
12: **end for**

---

**Training Details and Loss Behavior.** All diffusion models are fine-tuned for 1M iterations with batch size 256, Adam, and learning rate $10^{-4}$. For Min–Max training, we use EMA decay $\alpha = 0.9$ to smooth the group-wise losses used by the maximization step. Training is performed on 8 NVIDIA V100 GPUs (32GB) and takes roughly five hours per model. Table 6 reports losses normalized so that the smallest class loss for each model is equal to 1. Values closer to 1 therefore indicate more balanced training losses across groups. Conditional training strongly flattens the loss landscape, while reweighting and Min–Max also reduce loss gaps, with an effect that depends on the VP/VE parameterization.

*Table 6.* Normalized diffusion training losses on FFHQ. Values are averaged over noise levels and reported relative to the lowest class loss within each model; lower spread across columns indicates better loss balance.

| Model | Method | All | Male | Female |
|---|---|---|---|---|
| EDM-VP | Pretrain | 2.29±0.80 | 3.88±1.80 | 1.00±0.00 |
|  | Conditional | 1.02±0.02 | 1.00±0.00 | 1.03±0.03 |
|  | Reweighted | 1.03±0.03 | 1.00±0.00 | 1.05±0.05 |
|  | Min-Max | 1.04±0.04 | 1.00±0.00 | 1.07±0.08 |
| EDM-VE | Pretrain | 1.02±0.02 | 1.00±0.00 | 1.04±0.03 |
|  | Conditional | 1.02±0.01 | 1.00±0.00 | 1.03±0.02 |
|  | Reweighted | 1.02±0.02 | 1.00±0.00 | 1.04±0.03 |
|  | Min-Max | 1.02±0.01 | 1.00±0.00 | 1.03±0.02 |

Figure 5 complements the main-text VP loss plot with the corresponding EDM-VE curves. Unlike EDM-VP, the pretrained EDM-VE model is already relatively balanced across gender groups, which explains why the improvement margin is smaller for this architecture.

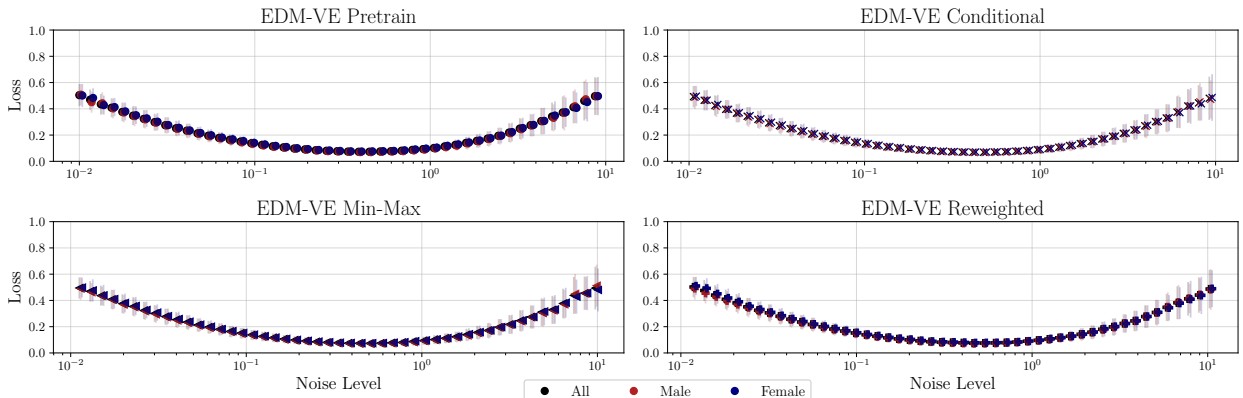

*Figure 5.* Estimated denoising loss by noise level for EDM-VE on FFHQ. Curves compare pretrained, reweighted, conditional, and Min–Max training.

### B.3.3. EVALUATING DIFFUSION MODELS

We evaluate fidelity and diversity using the Precision/Recall (P&R) framework of Kynkäänniemi et al. (2019), together with the Topological Precision & Recall (TopP&R) estimator of Kim et al. (2023a). Precision measures the fraction of generated features lying in the estimated real-data support, while recall measures the fraction of real features covered by generated samples. TopP&R makes this support estimation more stable by retaining statistically and topologically significant structures.

For FFHQ, features are extracted with a DINOv2 ViT (Oquab et al., 2024) fine-tuned on FFHQ. The same model is used as the oracle assigning male/female labels to generated samples. For numerical stability, features are projected with random Gaussian projections before running TopP&R. We repeat this procedure 10 times and report means, with standard deviations when relevant. Unless stated otherwise, each evaluation uses 20k generated samples per class and the same number of real samples per class. FID is reported for completeness; EGT conclusions rely on subgroup TopP&R precision, recall, and their sum.

**Results.** Table 7 provides the detailed FFHQ metrics for EDM-VP and EDM-VE under the evaluated sampling constraints. The compact disparity table used in the main text is Table 1.

*Table 7.* Detailed FFHQ evaluation metrics for EDM-VP and EDM-VE. Precision, recall, and $P + R$ are reported per group and as disparities; lower $\delta$ values indicate more balanced group-wise generation quality.

| Model | Gen | Method | Precision | | | | Recall | | | | FID | | | | $\delta$-PR |
|---|---|---|---|---|---|---|---|---|---|---|---|---|---|---|---|
| | | | All | Male | Female | $\delta$ | All | Male | Female | $\delta$ | All | Male | Female | $\delta$ | |
| VP | MGO | Pretrained | 95.2 | 95.2 | 97.2 | 2.0 | 84.3 | 91.7 | 87.5 | 4.2 | 2.33 | 3.12 | 2.78 | 0.34 | 2.2 |
| VP | MGO | Conditional | **98.0** | **96.8** | **98.4** | 1.6 | 80.2 | 91.5 | 84.7 | 6.8 | 2.75 | 3.41 | 3.32 | **0.08** | 5.2 |
| VP | MGO | Reweighted | 94.5 | 94.9 | 96.7 | 1.8 | 87.8 | 92.2 | 89.8 | 2.3 | 2.44 | 3.28 | 2.84 | 0.44 | **0.6** |
| VP | MGO | Min-Max | 89.4 | **96.8** | 96.7 | **0.1** | 91.2 | 86.6 | 86.4 | **0.2** | 2.75 | 3.51 | 3.27 | 0.24 | **0.3** |
| VP | EGO | Pretrained | 96.1 | 95.6 | 97.8 | 2.2 | 83.6 | 91.7 | 87.2 | 4.5 | 2.50 | 2.99 | 2.78 | 0.21 | 2.3 |
| VP | EGO | Conditional | 95.9 | **96.4** | **98.2** | 1.7 | 83.6 | **92.3** | 84.6 | 7.7 | 2.85 | 3.36 | 3.37 | **0.00** | 6.0 |
| VP | EGO | Reweighted | 93.6 | **96.1** | 96.4 | 0.3 | 86.9 | 90.8 | 89.9 | **1.0** | 2.62 | 3.25 | 2.89 | 0.36 | **0.6** |
| VP | EGO | Min-Max | 92.7 | **96.7** | 97.0 | **0.3** | 88.3 | 86.7 | 85.8 | **0.8** | 2.96 | 3.35 | 3.27 | **0.08** | **0.5** |
| VE | MGO | Pretrained | 97.9 | 95.9 | 98.0 | 2.1 | 81.4 | 87.8 | 83.4 | 4.4 | 2.49 | 3.27 | 2.92 | 0.35 | 2.3 |
| VE | MGO | Conditional | 95.6 | **98.2** | 97.4 | **0.9** | 86.4 | 87.1 | 84.4 | 2.7 | 2.88 | 3.54 | 3.43 | **0.11** | 3.5 |
| VE | MGO | Reweighted | 90.0 | 94.6 | 95.3 | **0.8** | 90.5 | 90.2 | 87.2 | **3.0** | 2.55 | **3.22** | 3.17 | **0.04** | 2.2 |
| VE | MGO | Min-Max | 89.2 | **96.9** | 97.2 | **0.3** | 91.4 | 89.5 | 88.4 | **1.1** | 2.81 | 3.67 | 3.34 | 0.33 | **0.8** |
| VE | EGO | Pretrained | 96.4 | 97.2 | 97.6 | 0.4 | 81.6 | 85.8 | 83.6 | 2.2 | 2.65 | 3.15 | 2.92 | 0.23 | 1.8 |
| VE | EGO | Conditional | **97.5** | 95.5 | **98.7** | 3.2 | 80.7 | **90.9** | 81.2 | 9.7 | 3.00 | 3.50 | 3.47 | **0.03** | 6.5 |
| VE | EGO | Reweighted | 93.2 | 94.7 | 94.7 | **0.0** | 88.7 | 89.8 | 90.4 | **0.5** | 2.77 | 3.20 | 3.25 | **0.05** | **0.5** |
| VE | EGO | Min-Max | 87.2 | **98.1** | 96.1 | 2.0 | 91.4 | 87.8 | 90.1 | 2.3 | 3.08 | 3.53 | 3.34 | 0.19 | **0.3** |

### B.3.4. GENERALITY ACROSS IMAGE DATASETS

The main text focuses on FFHQ because it offers a natural image-generation counterpart to the Wikipedia biography experiments: both settings evaluate generation quality across gender-defined groups. To check whether the conclusions depend on this particular dataset, we also evaluate the same EDM-VP and EDM-VE training strategies on CIFAR-10 and AFHQv2. These datasets keep the modality fixed while changing the semantic structure and the number of labels: FFHQ uses two gender groups, AFHQv2 uses three animal-domain labels, and CIFAR-10 uses ten object categories. The class labels in CIFAR-10 and AFHQv2 are not sensitive attributes; they are used here as controlled group labels to test whether Min–Max behavior remains stable when the number of groups changes.

*Table 8.* Additional diffusion results across image datasets. We report EGT disparities for precision, recall, their sum, and FID. All $\delta$-P, $\delta$-R, and $\delta$-PR values are in percentage points; $\delta$-FID is reported in FID units. Lower is better for all columns.

**FFHQ**

| Model | Method | $\delta$-P | $\delta$-R | $\delta$-PR | $\delta$-FID |
|---|---|---|---|---|---|
| VP | Pretrained | 2.0 | 4.2 | 2.2 | 0.34 |
| | Conditional | **1.6** | 6.8 | 5.2 | **0.08** |
| | Reweighted | 1.8 | 2.3 | 0.6 | 0.44 |
| | Min–Max | **0.1** | **0.2** | **0.3** | 0.24 |
| VE | Pretrained | 2.1 | 4.4 | 2.3 | 0.35 |
| | Conditional | 0.9 | 2.7 | 3.5 | **0.11** |
| | Reweighted | 0.8 | 3.0 | 2.2 | **0.04** |
| | Min–Max | **0.3** | **1.1** | **0.8** | 0.33 |

**CIFAR-10**

| Model | Method | $\delta$-P | $\delta$-R | $\delta$-PR | $\delta$-FID |
|---|---|---|---|---|---|
| VP | Pretrained | 6.9 | 8.9 | 4.5 | 4.86 |
| | Conditional | **6.3** | 10.7 | **4.0** | **4.08** |
| | Reweighted | **3.6** | 11.2 | 8.6 | 5.60 |
| | Min–Max | 6.0 | **8.4** | **3.6** | 4.12 |
| VE | Pretrained | 5.6 | 9.3 | 5.4 | 4.61 |
| | Conditional | 6.3 | **8.4** | **4.0** | **4.08** |
| | Reweighted | **2.8** | 8.9 | 7.7 | 5.73 |
| | Min–Max | 5.4 | **6.4** | **2.8** | 4.38 |

**AFHQv2**

| Model | Method | $\delta$-P | $\delta$-R | $\delta$-PR | $\delta$-FID |
|---|---|---|---|---|---|
| VP | Pretrained | 6.2 | 2.9 | 4.3 | 3.18 |
| | Conditional | **4.4** | 3.0 | **2.6** | **2.81** |
| | Reweighted | **3.9** | 3.1 | 7.0 | **3.07** |
| | Min–Max | 5.6 | **2.0** | **1.6** | **1.14** |
| VE | Pretrained | 3.9 | 2.9 | 3.7 | 3.20 |
| | Conditional | 4.6 | 4.0 | **1.6** | **2.70** |
| | Reweighted | 1.5 | 8.8 | 10.3 | 3.56 |
| | Min–Max | **1.5** | **2.4** | 1.9 | **1.99** |

Across these datasets, Min–Max remains the most consistent way of reducing the combined $\delta$-PR disparity. On FFHQ, it strongly improves both VP and VE models. On CIFAR-10, where the ten-way label structure makes the objective more demanding, Min–Max still achieves the best $\delta$-PR for both EDM variants. On AFHQv2, Min–Max again gives the strongest $\delta$-PR and $\delta$-FID on EDM-VP, and remains competitive on EDM-VE. These results support the main-text conclusion that explicitly targeting the worst group is more reliable than proportion-oriented baselines, while also showing that the magnitude of the gain depends on the dataset and label cardinality.

### B.3.5. GENERATED IMAGE SAMPLES

We additionally provide generated samples from each diffusion model and training strategy. In each grid, rows correspond to training methods and columns correspond to shared random seeds; the number below each image is the predicted group

label assigned by the dataset oracle. Sharing the same seeds across methods makes it easier to compare how the visual content changes with the training strategy, and to observe cases where visually similar samples differ mainly through their group label.

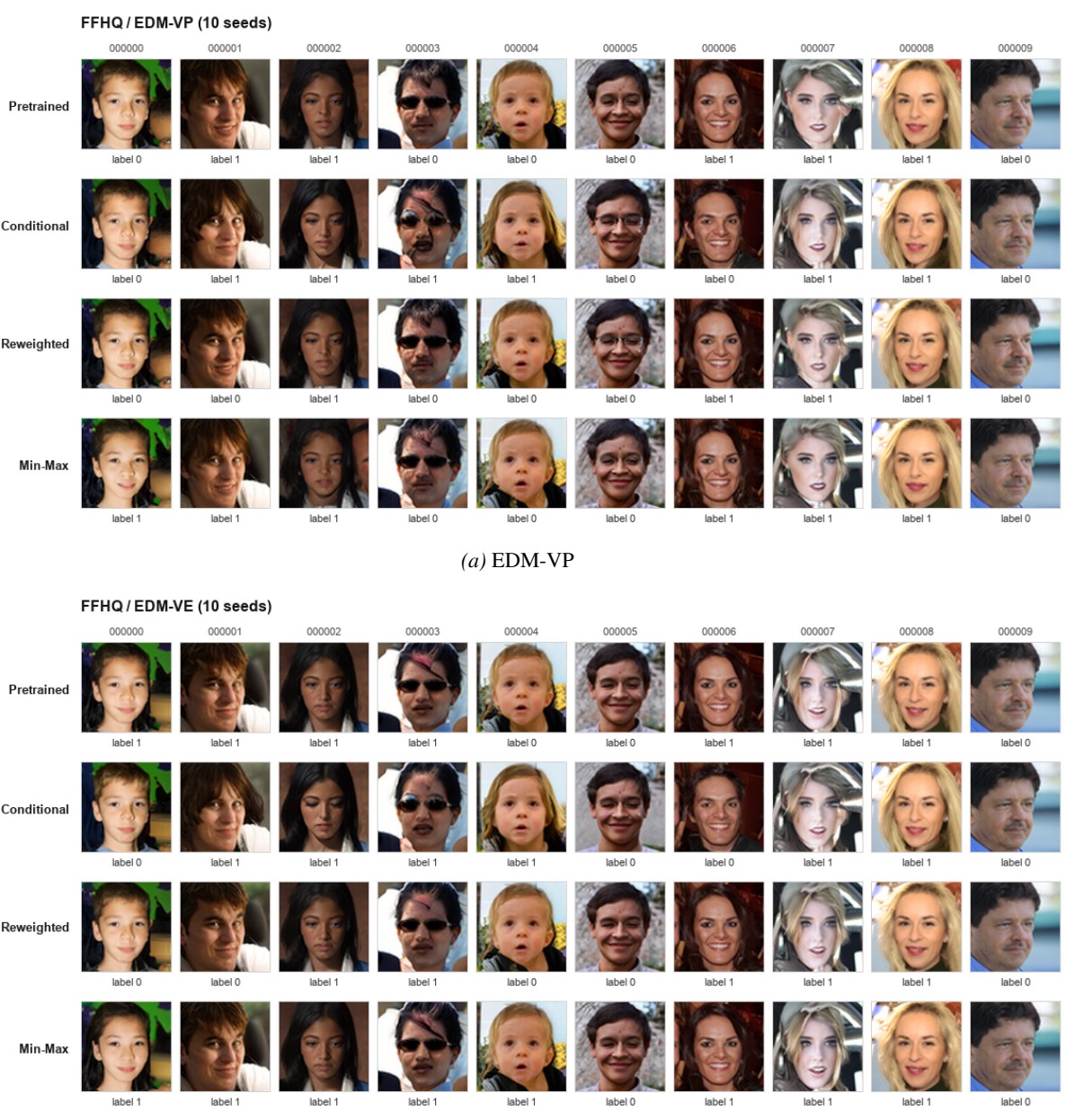

*(a)* EDM-VP

*(b)* EDM-VE

*Figure 6.* Generated FFHQ samples. Rows are training methods and columns are shared seeds; labels are shown below each image.

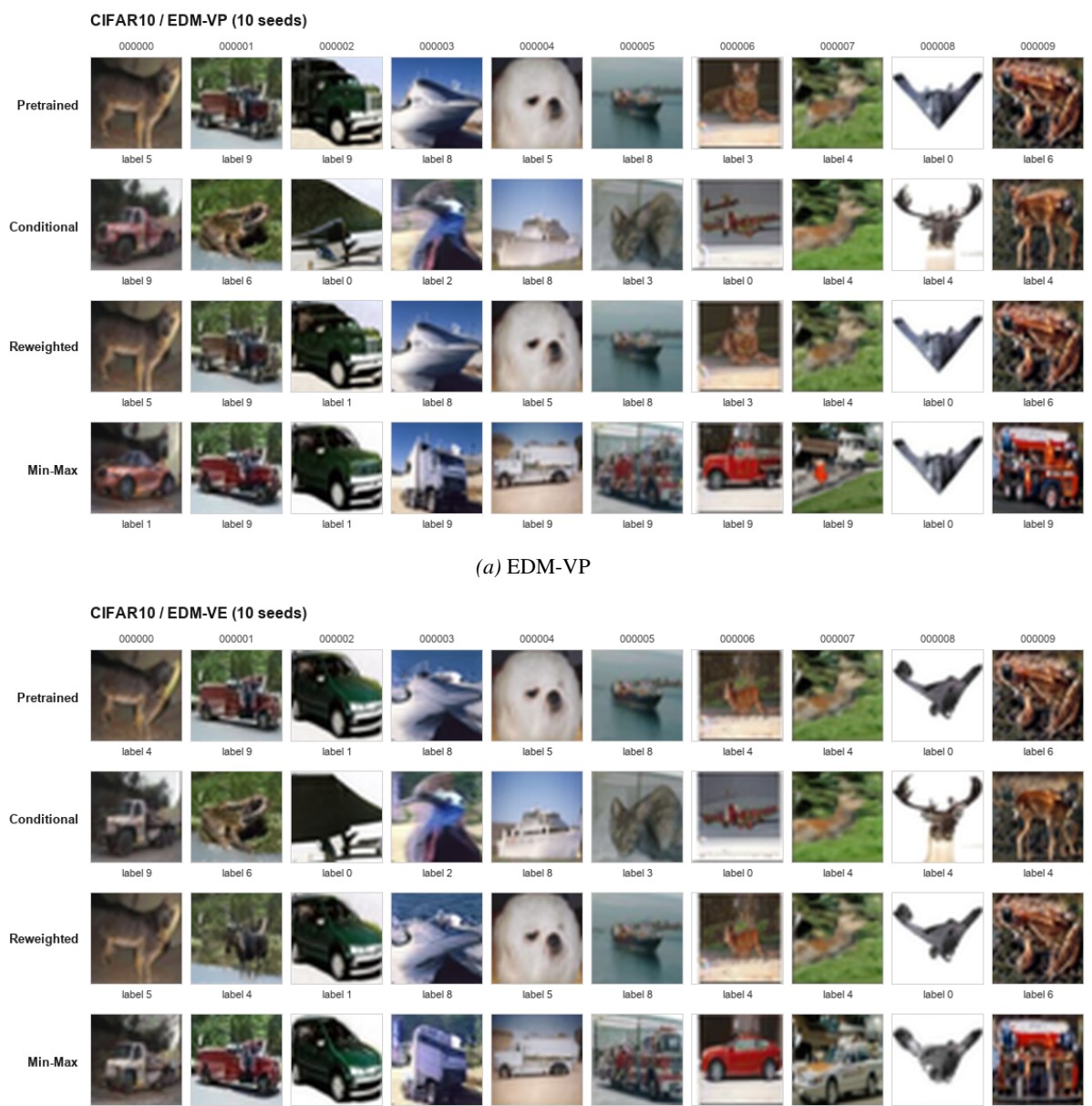

*(a)* EDM-VP

*(b)* EDM-VE

*Figure 7.* Generated CIFAR-10 samples. Rows are training methods and columns are shared seeds; labels are shown below each image.

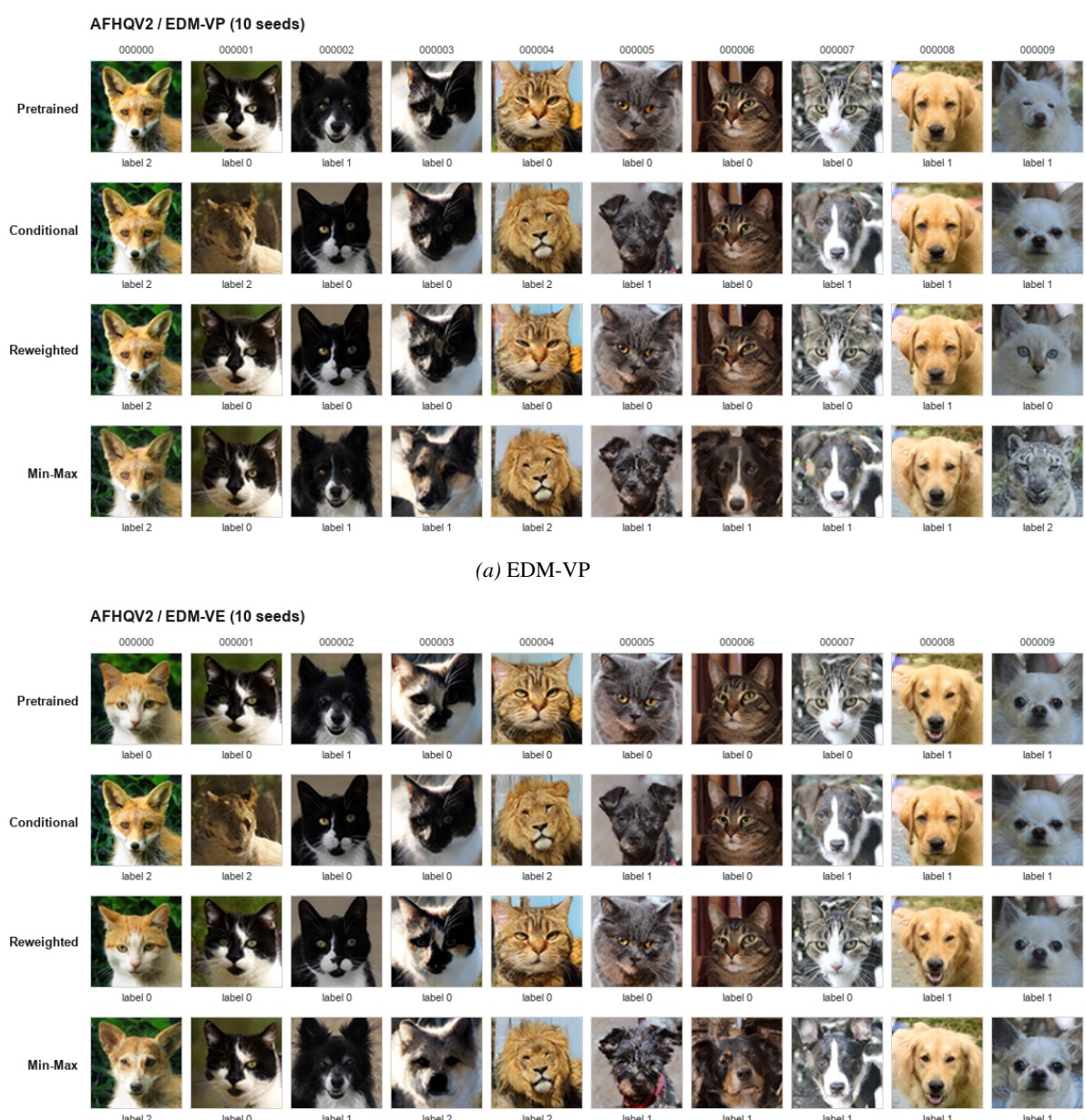

*(a)* EDM-VP

*(b)* EDM-VE

*Figure 8.* Generated AFHQv2 samples. Rows are training methods and columns are shared seeds; labels are shown below each image.

## B.4. Large Language Models

### B.4.1. TRAINING LARGE LANGUAGE MODELS

We fine-tune the chat variants of LLaMA 3.2 at two scales (**1B** and **3B** parameters) using parameter-efficient adapters (LoRA). Intuitively, LoRA inserts small trainable matrices into the attention/MLP layers so that we only update a tiny fraction of weights, which is more memory- and time-efficient than full fine-tuning.

**Prompting and Conditioning.** Fine-tuning uses a two-message chat template with a system instruction specifying a Wikipedia-style tone and a user query requesting a short biography summary. The model is trained to produce the response:

```
SYSTEM_PROMPT = """
```

```
You are a helpful assistant that writes Wikipedia-style biography
summaries of notable individuals.
Your writing should follow the tone, style, and structure of
Wikipedia "Summary" sections:
- Neutral, encyclopedic tone
- No personal opinions or promotional language
- Concise but informative, focusing on key life events, career,
and achievements
- Chronological ordering of major facts
- Use proper sentences, not bullet points
- The biography text should start with "Biography:" followed
directly by the text in the next line.
Do not invent references or citations. Do not include external
commentary about the writing task.
"""

CHAT_ZERO_SHOT = (
    'Write the "Summary" section of a Wikipedia article about
    a person of your choice.'
)

messages = [
    {"role": "system", "content": SYSTEM_PROMPT},
    {"role": "user",   "content": CHAT_ZERO_SHOT},
]
```

To obtain *conditional* generations, the same template is used while appending a short attribute constraint (here, gender) directly in the user message:

```
CHAT_ZERO_SHOT = (
    'Write the "Summary" section of a Wikipedia article about
    a person of your choice.'
)
if gender is not None:
    user_prompt = CHAT_ZERO_SHOT + f" It should be a biography
    of a {gender} person."
messages = [
    {"role": "system", "content": SYSTEM_PROMPT},
    {"role": "user",   "content": user_prompt},
]
```

This prompt-based conditioning makes it straightforward to control attribute proportions at inference time (e.g., 50/50 or 75/25), and directly matches the EGO/MGO controls discussed earlier.

**Training Details.** We fine-tune the models for **6 epochs** with a batch size of **64** and a learning rate of **1e-4** using the AdamW optimizer. We use a LoRA rank of **8**. Training is performed on a single NVIDIA H100 GPU with 80GB of memory and takes approximately **2 hours** for the 1B model and **5 hours** for the 3B model.

### B.4.2. EVALUATING LARGE LANGUAGE MODELS

We evaluate text generations with the same P&R formalism and TopP&R support estimation used for images, but in a *text-embedding space*. Concretely, we embed both real (Wikipedia) and generated biographies with a strong sentence-embedding model (we use a 4B-parameter open-source text-embedding model; reference omitted for anonymity). We then apply a random Gaussian projection to reduce dimensionality and run TopP&R to estimate topological supports and compute precision/recall. This process is repeated **10** times with independent projections; we report the mean (and, when relevant,

the standard deviation).

**Oracle for Sensitive Attribute.** For LLM outputs, we assign gender labels using a simple pronoun-based heuristic: we scan each biography and count occurrences of gendered terms (e.g., *he/him/his* vs. *she/her/hers*). The majority class is used as the predicted label; ties or ambiguous cases are left unassigned. While noisy, this heuristic works well on short Wikipedia-style summaries and avoids extra training.

**Notes.** As with images, we compute group-wise P&R (and P+R) under TopP&R on embeddings to quantify fidelity and coverage per sensitive group. We also report FID-like global metrics for completeness, but our fairness conclusions rely on group-wise TopP&R.

*Table 9.* Evaluation metrics for different LLMs on Wikipedia Bio dataset for the different methods. We plot the results for LLaMA-1B and LLaMA-3B models, as well as for Gemma-4B model pretrained and instruct-tuned.

| Model | Method | Precision | | | | Recall | | | | $\delta$PR |
| | | All | Male | Female | $\delta$ | All | Male | Female | $\delta$ | |
|---|---|---|---|---|---|---|---|---|---|---|
| LLaMA-3.2-Chat 1B | Pretrained | 50.45 | 55.68 | 76.86 | 21.18 | 86.08 | 77.03 | 65.57 | 11.46 | 32.65 |
| | Conditional | 45.85 | 55.68 | 74.76 | **19.09** | **86.29** | 74.18 | 66.36 | **7.81** | **26.90** |
| | Reweighted | **50.51** | 55.43 | 76.72 | 21.29 | 85.29 | 76.12 | **66.91** | 9.22 | **30.50** |
| | MinMax | **50.96** | 54.79 | 74.34 | **19.55** | 84.60 | **77.11** | 69.39 | **7.73** | 27.28 |
| LLaMA-3.2-Chat 3B | Pretrained | 51.77 | 53.06 | 78.43 | 25.37 | 91.37 | 87.40 | 72.34 | 15.06 | 40.43 |
| | Conditional | 46.48 | **53.14** | 77.44 | **24.31** | **93.67** | 85.38 | **73.11** | **12.26** | 36.57 |
| | Reweighted | 50.16 | 54.75 | 77.63 | **22.88** | 91.88 | 84.94 | 72.65 | 12.29 | 35.16 |
| | MinMax | 49.45 | **54.78** | 77.05 | **22.27** | 91.86 | 85.76 | 72.89 | 12.87 | **35.13** |
| Gemma-3 4B (pt) | Pretrained | 63.98 | 66.22 | 87.12 | 20.90 | 96.71 | 97.03 | 84.90 | 12.13 | 33.03 |
| | Conditional | 57.50 | 65.46 | **88.64** | 23.17 | **98.71** | **97.39** | **86.30** | **11.08** | 34.25 |
| | Reweighted | 62.41 | **67.79** | 87.47 | **19.67** | 97.27 | 96.68 | 84.97 | 11.71 | **31.38** |
| | MinMax | **64.24** | **67.98** | 87.11 | **19.13** | 97.24 | 97.36 | 85.70 | 11.67 | **30.79** |
| Gemma-3 4B (it) | Pretrained | 60.31 | 62.59 | 84.89 | 22.30 | 97.18 | 96.78 | 90.28 | 6.50 | 28.81 |
| | Conditional | 55.58 | 61.68 | **85.85** | 24.16 | **98.21** | 96.72 | 88.31 | 8.41 | 32.57 |
| | Reweighted | 59.83 | **64.55** | **85.47** | **20.91** | 97.52 | 96.07 | 89.87 | **6.20** | **27.11** |
| | MinMax | 59.10 | **63.97** | **85.62** | **21.65** | 97.85 | 96.08 | 89.79 | 6.28 | **27.93** |

### B.4.3. WIKIPEDIA BIOGRAPHIES DATASET

The Wikipedia Biographies dataset is derived from the corpus introduced by Bronnec et al. (2024), which contains biography headers of individuals. Below, we present examples of biographies generated by the different models.

**Unconditional Generation:**

| Model | Method | Text | Gender |
|---|---|---|---|
| 1b | Pretrained | Joseph Louis Casimir Czerniak (Polish: [jzef luiz ksnak]; 15 February 1898 14 March 1943) was a Polish politician who served as the 20th Prime Minister of Poland from 1925 to 1928 and again from 1935 to 1939. He was the leader of the Polish Socialist Party, a Polish political party founded in 1919, and served as its leader from 1924 to 1935. He was the first Polish Prime Minister to be elected to the Polish Sejm. He was assassinated in 1943. Czerniak was born in Warsaw. He studied at the Warsaw University, graduating in 1921. He was a member of the Polish Socialist Party and participated in the PolishSoviet War. In 1925, he was appointed Prime Minister after the resignation of Stanisaw Wojciech Mikoaj Gombrzyski. He was re-elected in 1928, but was dismissed by the Sejm in 1935. He was assassinated in 1943. | male |
| 1b | Pretrained | Robert Charles Stevenson (29 June 1847 14 September 1927) was an English actor, comedian and comedian who appeared in numerous pantomimes and musical comedies, and was best known for his starring roles in the 1912 film "The Tragedy of Esther Waters" and the 1915 film "The Homecoming of Robin Hood". He was born in London and began his career as a professional actor in the West End, and later moved to the United States, where he appeared in a number of stage shows and films. He died in 1927. | male |

| 1b | Min-Max | Joseph Louis Broussard (November 19, 1881 July 8, 1962) was a Major League Baseball pitcher who played for the St. Louis Cardinals, Philadelphia Athletics, New York Yankees, and Chicago Cubs. He was a member of the Cardinals' 1922 World Series championship team and was a member of the 1926 National League pennant winning team. He was the first pitcher to throw a no-hitter in the National League, and was the first pitcher to throw a complete game shutout in the National League, accomplishing the feat in 1928 while pitching for the Yankees. Born in El Paso, Texas, Broussard began his baseball career with the Tucson Toros of the Pacific Coast League. He was signed by the Cardinals in 1909 and made his major league debut in 1910. He was traded to the Athletics in 1912, where he was released after the 1914 season. He returned to the Cardinals in 1915 and pitched for the team until his retirement in 1925. Broussard was inducted into the National Baseball Hall of Fame in 1962. | male |
|---|---|---|---|
| 1b | Min-Max | Robert "Bob" C. Henningsen (born June 7, 1935) is an American retired professional baseball player who played in Major League Baseball for 16 seasons as a catcher. He played his entire career with the St. Louis Cardinals, and is a member of the Cardinals' franchise's Hall of Fame. Henningsen was a key player on the Cardinals' 1962 World Series championship team and was a three-time All-Star. He was a member of the Cardinals' 1963 National League pennant team. He also won a World Series championship in 1964 with the Cardinals and was named the World Series MVP. | male |
| 1b | RW | Joseph Louis Casimir Czerniak (Polish: [jzef luiz ksnak]; 15 February 1898 14 March 1943) was a Polish politician who served as the 20th Prime Minister of Poland from 1925 to 1928 and again from 1935 to 1939. He was the leader of the Polish Socialist Party, a Polish political party founded in 1919, and served as its leader from 1924 to 1935. After the PolishSoviet War, he became a key figure in the Polish government and was appointed Prime Minister in 1935. He was the first Polish government leader to be imprisoned by the Nazis. He died in 1943 while in custody. Czerniak was born in the city of Krakw. He studied law at the University of Krakw, where he became involved in the Polish Socialist Party. He was a member of the Polish government in exile from 1925 to 1928, and was a key figure in the Polish government of 1935. He was imprisoned by the Nazis after the invasion of Poland in 1939 and died in custody in 1943. | male |
| 1b | RW | Robert Charles Stevenson (29 November 1847 14 September 1927) was an English actor, comedian and comedian who appeared in numerous pantomimes and musical comedies, and was best known for his starring roles in the 1912 film "The Tragedy of Esther Waters" and the 1915 film "The Homecoming of Robin Hood". He was born in London and educated at Harrow School and the Royal Academy of Dramatic Art. He began his acting career in London's West End, performing in a number of pantomimes and musical comedies, before moving to the United States, where he became a popular actor and comedian in the early 20th century. He appeared in a number of films, and toured the United States in the 1910s and 1920s, performing in vaudeville and revues. He died in 1927. | male |
| 3b | Pretrained | Joseph "Joe" John "Bull" Connolly (March 2, 1895 November 7, 1969) was an American professional baseball player who played in Major League Baseball (MLB) for 17 seasons, all with the New York Yankees. He was a left-handed pitcher and a member of the Yankees' 1923 World Series championship team. Connolly was the Yankees' ace pitcher in the 1923 season, posting a 207 winloss record with a 2.61 earned run average (ERA). He was also a key member of the Yankees' 1927 team that won the World Series, and pitched in 18 games for the team. Connolly was a member of the Yankees' rotation in the early 1920s, but his career was marked by inconsistency and a series of injuries. He was a member of the Yankees' starting rotation for only 18 of his 17 seasons in the major leagues, and he was out of baseball for several years due to various injuries. He was released by the Yankees in 1930, but returned to the team in 1934. He played for the Yankees until 1938, when he retired. | male |
| 3b | Pretrained | Charles Charles Victor Sargent (March 21, 1841 May 25, 1927) was an American landscape gardener, horticulturist, and botanist. He was a professor of horticulture at the Massachusetts Agricultural College (now the University of Massachusetts Amherst) for over 40 years and was the first president of the American Horticultural Society. He is best known for his work on the development of the Boston Public Garden and the Boston botanical garden at Harvard University. Sargent was born in Boston, Massachusetts, and grew up in a family of modest means. He was apprenticed to a nurseryman at the age of 15, and after completing his apprenticeship, he worked in several other gardens in the United States. In 1872, he was appointed as a professor of horticulture at the Massachusetts Agricultural College, and he remained there until his retirement in 1913. Sargent was a prolific writer, and his books on horticulture were widely read in the United States and abroad. He was also a prolific writer on botanical topics, and his works were published in numerous journals and magazines. | male |

| 3b | Min-Max | Joseph "Jack" Gorman (born 10 November 1985) is a former English professional footballer who played as a striker for several clubs, including Wigan Athletic, Ipswich Town and Crawley Town. Gorman began his career at Wigan Athletic, signing for the club in 2005 from Manchester City. He made his debut for the club in 2005 and scored one goal in 14 appearances. He was released from the club at the end of the 200506 season and joined Ipswich Town in 2006. He spent two years with the club, scoring six goals in 51 appearances before moving to Crawley Town in 2008. He spent two seasons with the club, scoring 21 goals in 69 appearances before leaving the club in 2010. | male |
| 3b | Min-Max | Charles William Williams (6 July 1831 7 January 1910) was an English-born Australian cricketer who played first-class cricket for the New South Wales cricket team. He was a right-handed batsman and occasional wicket-keeper. Born in England, Williams moved to Australia with his family when he was eight years old. He was educated at the Sydney Church of England Grammar School and began his cricketing career with the school team, before making his first-class debut for New South Wales in 1853. He played for the team for the next 12 years, and was a member of the 185960 team that toured England, where he also played for the Marylebone Cricket Club. He was known for his batting, particularly his ability to score big centuries. He played in the first ever Test match in 1877 and was also a member of the New South Wales team that toured England in 188283. He retired from first-class cricket in 1865, but continued to play in minor league cricket until his death in 1910. | male |
| 3b | RW | Joseph "Joe" O'Connell (May 19, 1920 August 24, 1997) was an American politician who served as a member of the New Jersey General Assembly from the 2nd district from 1958 to 1964 and from 1967 to 1971. He was a Democrat. O'Connell was born in New York City and graduated from the New York City College of Pharmacy. He served in the United States Army during World War II and was awarded the Purple Heart. He graduated from Rutgers University and later earned a master's degree in education from Columbia University. He was a teacher and a pharmacist before entering politics. He was elected to the New Jersey General Assembly in 1957 and served until 1964. He was defeated for re-election in 1964, and did not seek election again until 1966. He served another term until 1971, when he was defeated in a Republican primary. | male |
| 3b | RW | Charles Francis "Charlie" Jones (February 27, 1886 January 15, 1968) was an American baseball player and manager in Major League Baseball. He played in Major League Baseball from 1909 to 1925, playing for the Philadelphia Athletics, New York Giants, and Boston Braves. He also managed the St. Louis Browns from 1928 to 1929. Jones was a four-time All-Star and won the National League batting title in 1912. He is a member of the National Baseball Hall of Fame. He is the only player in Major League Baseball history to have won a batting title, a Gold Glove Award, and a World Series championship. | male |

## Conditional Generation:

| Model | Method | Text | Gender |
|-------|--------|------|--------|
| 1b | Pretrained | Vivian Joseph O'Neal (born April 21, 1987) is an American professional baseball pitcher who played in Major League Baseball (MLB) as a relief pitcher for the New York Yankees. A former high school standout, he played college baseball for the University of Michigan Wolverines. He was drafted by the Yankees with the 8th round pick of the 2006 MLB draft. O'Neal was named the Yankees' Minor League Pitching Prospect of the Year in 2008. He was named a Golden Glove winner for his defensive play at shortstop in 2011. He was traded to the Los Angeles Angels in 2012. O'Neal was released by the Angels in 2013 and signed with the Baltimore Orioles in 2014. He was traded to the Kansas City Royals in 2015. He was traded to the Texas Rangers in 2016. He was released by the Rangers in 2017 and signed with the New York Yankees in 2018. He was traded to the Miami Marlins in 2019. He was released by the Marlins in 2020 and signed with the Seattle Mariners in 2021. | male |

| | | | |
|---|---|---|---|
| 1b | Pretrained | Lucius Julius Caesar (c. 52 BC 44 BC) was a Roman general and statesman who served as dictator of Rome in 49 BC. He was a member of the Julian clan and a member of the Julian clan, a patrician family that played a significant role in Roman politics. Caesar was the son of Gaius Julius Caesar and Cornelia, a member of the Julian clan. He was appointed as a quaestor in 43 BC, and served in the Roman Senate in 42 BC, but was expelled from the Senate in 41 BC after he was accused of treason and exiled to the island of Corsica. He returned to Rome in 40 BC, and served as a quaestor in 39 BC. He was appointed dictator of Rome in 49 BC, and was elected dictator for the fourth time in 44 BC, becoming the first Roman dictator since 49 BC. He was assassinated in 44 BC. Caesar's reign was marked by a series of military victories against the Gauls, and he was celebrated for his military campaigns, which included the conquest of the Roman province of Hispania, the conquest of the island of Sicily, and the expulsion of the Parthian Empire from the Roman province of Cappadocia. He also reformed the Roman calendar, introducing the Julian calendar, and reformed the Roman Senate, establishing the office of the praetor maximus. | male |
| 1b | Min-Max | Vivian James (ne Wilson; 4 October 1919 28 December 1994) was an English professional golfer who competed in the 1947 British Open and the 1951 US Open. She was the first woman to win a major championship. She was also the first woman to be ranked number one in the world golf rankings for a period of two years. She was a member of the first class of golfers to be inducted into the World Golf Hall of Fame. | female |
| 1b | Min-Max | Lucius Marcellus Valerius Rufus (fl. 1st century BC) was a Roman senator and statesman from the gens Rufi, a patrician family that included several prominent politicians and military leaders in the Roman Republic. He was a member of the first century BC, and was a member of the Curia of the Roman Senate from 41 BC until his death in 1 BC. Rufus was known for his political and military career in the late Republic, during which he served as a general under the dictator Julius Caesar. He was appointed consul in 41 BC and served as governor of the province of Asia in 39 BC. He died in 1 BC. | male |
| 1b | RW | Vivian Vess, ne (born 1956) is a Canadian actress, best known for her roles in the TV series The Shield and its spin-off series Alphas. She was born in New York City and began her career in television and film in the early 1980s. She began as a child actress on the soap opera The Young and the Restless and later appeared on the television series Perfect Strangers, The Love Boat, and The Golden Girls. She gained recognition for her roles in the television series The Shield and its spin-off series Alphas, for which she won a Canadian Screen Award for Best Supporting Actress in a Drama Series. | female |
| 1b | RW | Lucius Quinctius Cincinnatus (c. 180c. 90 BCE) was a Roman statesman and military leader of the late Roman Republic. He was a member of the noble family of the Cincii, and was a prominent figure in the late Roman Republic, serving as a censor, quaestor, and tribune of the plebs. Cincinnatus was born in 180 BCE. He was a veteran of the Second Punic War, and served as a quaestor in the late 2nd century BCE. He was elected as a censors in 247 BCE, and then as a tribune of the plebs in 229 BCE. He served as censor again in 219 BCE, and was elected as a consul in 216 BCE. He was appointed as dictator in 214 BCE, and then as dictator again in 211 BCE. He was assassinated in 211 BCE, and succeeded by his nephew, Lucius Quinctius Calvus. | male |
| 3b | Pretrained | Joseph "Joe" O'Connell (May 19, 1920 August 4, 1997) was an American politician who served as a member of the New Jersey General Assembly from the 9th district from 1958 to 1964 and from 1967 to 1971. He was a Democrat. O'Connell was born in New York City and graduated from the New York City High School of Commerce. He served in the United States Army during World War II and the Korean War, and worked as a laborer and a truck driver. He was elected to the General Assembly in 1957 and served until 1964. He was again elected in 1966 and served until 1971. | male |
| 3b | Pretrained | Robert Charles "Bob" Jones III (born March 30, 1940) is an American businessman, politician, and Christian minister. He is best known for founding Liberty University, a Christian university in Lynchburg, Virginia, and for his advocacy of Christian nationalism and conservative politics. He was the chairman of the Board of Directors of the Family Research Council, a conservative Christian advocacy group, from 1982 until 2012. He was also the founder and chairman of the World Relief and Development Council, a Christian charity. Jones founded Liberty University in 1971 as a small Bible college in Lynchburg, Virginia, and expanded it into a large university by the 1980s. In the 1980s, he was a prominent figure in the Christian Right, and he ran for the Republican presidential nomination in 1988. He has been a vocal critic of the separation of church and state and has advocated for the teaching of creationism in public schools. He has been described as a "Christian fundamentalist" and a "evangelical". | male |

| | | | |
|---|---|---|---|
| 3b | Min-Max | Joseph "Joe" Jones (February 28, 1897 December 23, 1971) was an American professional baseball player who played for the Philadelphia Athletics, Chicago Cubs, and Brooklyn Dodgers of Major League Baseball (MLB). He played as a pitcher and outfielder. Jones was born in New York City and attended George Washington University, where he played college baseball for the George Washington Pioneers. After college, he was signed by the Athletics and made his MLB debut in 1918. He played for the Athletics for eight seasons, including three All-Star appearances, and was a member of the 1927 World Series team. He was traded to the Cubs in 1929, where he played for two seasons, and was then traded to the Dodgers in 1931. He played for the Dodgers for four seasons before his MLB career was cut short by a series of injuries. | male |
| 3b | Min-Max | Robert James "Bobby" Jones (23 July 1890 2 September 1971) was an English professional footballer who played as a centre-half. He made over 200 appearances for the first team of Southampton Football Club, and also played for the England national team. Born in Southampton, Jones began his career with local side East End United before joining Southampton in 1908. He made his first-team debut in 1910 and became a regular player for the club, helping them win the Southern League title in 191213. He also played for the England national team, earning 14 caps, and was part of the team that won the 1920 Summer Olympics. Jones moved to Fulham in 1914, but returned to Southampton in 1919, playing until his retirement in 1925. He later managed the club's youth team and was appointed chairman of the club in 1954. He was knighted in 1958 for his services to football. | male |
| 3b | RW | Joseph "Joe" Lawler (born 2 January 1968) is an English former professional footballer who played as a midfielder for several clubs including Manchester City, Ipswich Town and Sunderland. He also had a brief spell at Middlesbrough. Lawler began his career with his hometown club Manchester City, before moving to Ipswich Town in 1990. He won the First Division title in his first season at the club, and played in the 1992 FA Cup final. He moved to Sunderland in 1993, where he won the First Division title again, and played in the 1994 FA Cup final. He moved to Middlesbrough in 1995, but left after just one season. He had a brief spell at Norwich City in 1997 before retiring. | male |
| 3b | RW | Robert Francis "Bob" Johnson (January 23, 1894 August 7, 1971) was an American professional baseball player. He played in Major League Baseball (MLB) as a pitcher for the St. Louis Browns from 1913 to 1918. He was a left-handed thrower and batted and threw right-handed. Johnson was a member of the 1915 World Series championship team. He is best known for throwing a no-hitter in Game 1 of the 1915 World Series, and for his 1916 season in which he won 13 games and lost just two, and was named the American League leader in shutouts with 11. Johnson was also the American League leader in shutouts with 11 in 1917, and he was a member of the St. Louis Browns' 1918 World Series championship team. Johnson's career was cut short by an injury, as he suffered a shoulder injury in 1918 and was forced to retire from baseball. He later worked as a baseball scout for the Chicago Cubs and was also involved in the promotion of the St. Louis Browns. | male |

