# OpenReview forum: "Equalized Generative Treatment: Matching f-divergences for Fairness in Generative Models"
_ICML.cc/2026/Conference — ICML 2026 regular_

### Official Review · Reviewer_GDKT · 2026-02-15

**Soundness:** 2
**Presentation:** 1
**Significance:** 2
**Originality:** 3
**Overall Recommendation:** 4
**Confidence:** 3

**Summary:**

The authors propose EGT and a min-max fine-tuning method to address the different generation quality across attributes, and evaluate their effectiveness on two tasks. The presentation of this paper is difficult to understand. The method may have some theoretical application value, but the experiments do not demonstrate it. The main problem of this paper is that the experiments are conducted only on gender.

**Compliance With Llm Reviewing Policy:**

Affirmed.

**Final Justification:**

The authors addressed most of my concerns during the two rebuttal rounds, and further demonstrated the generality of the proposed method through additional experiments. Therefore, I increased my score from 2 to 3, and then further to 4. I believe the authors have shown a responsible attitude toward their work.

**Key Questions For Authors:**

Refer to question 1 2 6 7 8 of Strengths And Weaknesses.

**Limitations:**

This paper relies on additional labels of sensitive attributes, which may make the method inapplicable to tasks without such labels, such as text-to-image generation using natural language.

**Strengths And Weaknesses:**

Soundness

1. The min-max method seems likely to reduce the quality of high-quality generations in order to achieve fairer generation. I think it makes more sense to specifically optimize low-quality generations.

2. The experiments evaluate gender as the sensitive attribute, which is insufficient. Since the sensitive attribute only serves as an additional label, any classification dataset could be used, such as large-scale ImageNet or very small CIFAR10.

Presentation

3. The writing logic is too difficult to understand. In my view, to address different generation quality across groups, the paper proposes Definition 4.1 based on Definition 2.1, and optimizes Definition 4.1 using the last formula on page 6. If the authors first introduce their main contribution and then describe the existing problem it solves, it would be much clearer.

4. The paper does not number the equations.

5. The Impact Statement section is missing.

Significance

6. The paper uses a 2025 4B text model for evaluation. Why does it only use the 2022 EDM for image generation?

7. The paper only includes ablation studies and does not compare with state-of-the-art related work.

8. The authors use extensive theory to demonstrate the shortcomings of existing methods. However, as this is a generation task, why not directly visualize the deficiencies of generated results in the manuscript?

---

> ### Author Rebuttal · Authors · 2026-03-31
>
> We thank the reviewer for their time and feedback. We believe that many of the perceived weaknesses arise from a misunderstanding of our main contributions and claims. We provide detailed responses to the raised points below.
>
> - (1) The min–max procedure is not intended to reduce the quality of high-quality generations; rather, it targets improving low-quality generations. Specifically, we minimize the loss for the class with the largest (worst) value. This aligns with your suggestion to optimize low-quality generations, now supported by our theoretical framework. We believe the agreement between intuition and theory provides a useful consistency check.
> - (2) We agree that other datasets, such as CIFAR-10 or ImageNet, could have been used in these experiments. However, the focus of our paper is on the generality of both the theoretical fairness definitions and the effectiveness of the Min–Max procedure across modalities. For this reason, we specifically selected a sensitive attribute that could be consistently evaluated in both image and text experiments.
>
> - (6) Although the EDM models were designed in 2022, they remain close to the state-of-the-art for generating FFHQ images at 64×64 resolution. Moreover, they are widely used in the diffusion model community for demonstrating theoretical results. Therefore, we believe that using these models should be viewed as a continuation of the state-of-the-art in evaluating image generation, rather than as a weakness.
>
> - (7) We appreciate the reviewer’s comment. We would like to clarify that Tables 1 and 2 include explicit comparisons between our approach and state-of-the-art methods. We hope that revisiting these sections will provide a clearer view of our contributions and help contextualize our results within the existing literature.
>
> - (8) Thank you for raising this point. We would be happy to include generated images from each model in the appendix. In particular, we observe that different models can produce visually similar samples that differ mostly only in sensitive attributes (e.g., very similar faces with different perceived gender). We already provide examples of generated text in the appendix, and we will extend this to include image samples as well.
>
> We also take all your other suggestion into account in subsequent revision.

---

> > ### Author Rebuttal · Reviewer_GDKT · 2026-04-02
> >
> > I think that evaluating on only two datasets cannot demonstrate the generality of the method. If the authors can provide additional experiments, I may increase my score again.

---

> > > ### Author Response · Authors · 2026-04-08
> > >
> > > Thank you for the follow-up comment. To further address the concern regarding generality, we added additional experiments on **CIFAR-10** [1] and **AFHQv2** [2], beyond the original two datasets.
> > >
> > > The goal of these new experiments is not to introduce a new fairness setting, but to test whether the same min-max training principle remains effective on other multi-class image generation benchmarks with different semantic structure and difficulty. In particular, these experiments allow us to verify that the behavior observed in our main results is not specific to the original datasets.
> > >
> > > For **CIFAR-10**, the dataset is already class-balanced, so the notions corresponding to MGO and EGO coincide in this setting. For this reason, we only report the MGO experiments. We compare the standard pretrained model, a reweighting baseline, a conditional baseline, and our min-max variant.
> > >
> > > For **AFHQv2**, we evaluate the same family of methods on a qualitatively different dataset with three animal categories, which provides an additional testbed with larger inter-class visual differences than gender-based partitions.
> > >
> > > Across both datasets, the min-max strategy remains competitive and generally improves the group-discrepancy metrics, especially $\delta$-FID and $\delta$-PR, supporting the claim that the approach is not tied to a single dataset or a single type of sensitive partition. We agree that broader empirical validation is valuable, and these added experiments are intended precisely to strengthen the empirical evidence for the generality of the method.
> > >
> > > [1] Learning Multiple Layers of Features from Tiny Images, Krizhevsky
> > >
> > > [2] StarGAN v2: Diverse Image Synthesis for Multiple Domains, Choi et al.
> > >
> > > ---
> > >
> > > ### CIFAR-10 (EDM-VE)
> > >
> > > | Train      | $\delta$-Precision | $\delta$-Recall | $\delta$-PR | $\delta$-FID |
> > > |------------|-------------------:|----------------:|------------:|-------------:|
> > > | Pretrained | 0.0555             | 0.0928          | 0.0542      | 4.6099       |
> > > | RW         | **0.0280**         | **0.0892**      | 0.0771      | 5.7328       |
> > > | Conditional | 0.0629             | **0.0843**      | **0.0399**  | **4.0818**   |
> > > | Min-Max    | **0.0536**         | **0.0639**      | **0.0275**  | **4.3787**   |
> > >
> > >
> > > ---
> > >
> > > ### CIFAR-10 (EDM-VP)
> > >
> > > | Train      | $\delta$-Precision | $\delta$-Recall | $\delta$-PR | $\delta$-FID |
> > > |------------|-------------------:|----------------:|------------:|-------------:|
> > > | Pretrained | 0.0690             | 0.0894          | 0.0445      | 4.8562       |
> > > | RW         | **0.0362**         | 0.1124          | 0.0856      | 5.5966       |
> > > | Conditional | **0.0629**         | 0.1073     | **0.0399**  | **4.0818**   |
> > > | Min-Max    |  **0.0602**             | **0.0843**           | **0.0359**  | **4.1176**   |
> > >
> > > ---
> > >
> > > ### AFHQv2 (EDM-VP)
> > >
> > > | Train      | $\delta$-Precision | $\delta$-Recall | $\delta$-PR | $\delta$-FID |
> > > |------------|-------------------:|----------------:|------------:|-------------:|
> > > | Pretrained | 0.0622             | 0.0292          | 0.0430      | 3.1806       |
> > > | RW         | **0.0394**         | 0.0307          | 0.0701      | **3.0678**   |
> > > | Conditional | **0.0435**         | 0.0295          | **0.0256**  | **2.8136**   |
> > > | Min-Max    | **0.0556**         | **0.0198**          | **0.0158**  | **1.1369**   |
> > >
> > > ---
> > >
> > > ### AFHQv2 (EDM-VE)
> > >
> > > | Train      | $\delta$-Precision | $\delta$-Recall | $\delta$-PR | $\delta$-FID |
> > > |------------|-------------------:|----------------:|------------:|-------------:|
> > > | Pretrained | 0.0389             | 0.0285          | 0.0373      | 3.2013       |
> > > | RW         | **0.0153**         | 0.0878          | 0.1031      | 3.5575       |
> > > | Conditional | 0.0456             | 0.0398          | **0.0158**  | **2.7020**   |
> > > | Min-Max    | **0.0152**         | **0.0243**          | **0.0189**  | **1.9862**   |

---

### Official Review · Reviewer_vNZS · 2026-02-18

**Soundness:** 2
**Presentation:** 1
**Significance:** 3
**Originality:** 3
**Overall Recommendation:** 4
**Confidence:** 3

**Summary:**

The paper suggests a new fairness metric for generative models. While prior work focused on equal representation for different groups in the generated samples, the paper proposes to also observe the difference between the sample quality for each group, called equalized generative treatment (EGT). The paper proves that common fairness definition may lead to bias in the generative quality for different groups, and proposes a training algorithm that indirectly minimizes EGT violations. The authors then compare different bias mitigation techniques on different modalities for empirical evidence of their theory.

**Compliance With Llm Reviewing Policy:**

Affirmed.

**Final Justification:**

The authors clarified my biggest concerns, and provided a detailed explanation of how they will edit the paper to make the paper more accurate and accessible. For this reason I increased my score from 3 to 4.

**Key Questions For Authors:**

1. Models that accurately capture the true distribution should have high precision and recall and low $f$-divergence. However, the authors report precision and recall as they should be minimized. Can the authors explain this relationship between precision recall and $f$-divergences, and include it in the paper? Without this explanation, the results in Tables 1-2 may make the reader think that the min-max method has almost 0 precision and recall, which is the worst result.
2. Definition 2.2 is incomplete. Should the second $\pi^{Q}_{a}$ be $\pi ^{P} _{a}$? Also, the definition of $P$ should be included here as well. Otherwise any $Q$ is $\delta$-EGO, because $P$ could be any distribution.
3. Could the authors explain Figure 1? In Fig 1a, the color choices make it difficult to distinguish between the curves, and the purpose of the blue $a=0$ and red $a=1$ is unspecified. In Fig 1b-c, what do the colors represent?
4. In the synthetic experiment presented in Fig 1, lines 191-192 don't explain how the Gaussians were fitted. Where they initialized with different parameters? How were they optimized? Also, the authors chose a level set of $D_{\text{JS}}=1$, yet the Jensen-Shannon divergence is upper-bounded by $1$, making this choice a somewhat misrepresenting example. How would the results look for a smaller fixed level $\epsilon$?
5. Definition 4.2 contains some editing errors ("the set of distributions for which" repeats twice, "for ever" instead of "for every"). As this definition is overloaded with notation, these errors put the other notations in question about the validity of the definition, as it implies the notations may be also incorrect. Additionally, what is the purpose of the definition? How does it solve the problem of not having a separate model for each population group?
6. Can the authors explain the conclusion of Theorem 4.3? The authors claim that "This result highlights the fact that enforcing EGT promotes the minimization of the highest $f$-divergence between conditionals among group", but the logical steps are missing. This conclusion may be true only if the bound is tight, which it may not be, or if the group $f$-divergence was the upper bound rather than lower bound.
7. How do the authors implement the min-max objective? The objective requires computing the $f$-divergence between $P_{a}$ and $Q_{a}$ for all groups. But the authors do not specify which $f$-divergence they used and how they computed it.
8. Can the authors include more technical details for the methods they used? The brief explanation in lines 348-370 is very high-level and ignores important technical details.
9. Tables 1-2 report values of $\delta$-P(recision), $\delta$-R(ecall) and other $\delta$- metrics, yet they are not defined. What is the meaning of adding $\delta$- before the metric name? Furthermore, it should be noted explicitly that P and R stand for Precision and Recall.

**Limitations:**

The paper is missing the required impact statement.

**Strengths And Weaknesses:**

## Strengths
1. The paper formalizes an important metric for fairness in generative models that may be overlooked by prior works.
2. The paper reports results over different modalities, algorithms and metrics.

## Weaknesses
1. Some definitions are not detailed enough, and some definitions are over-detailed and confusing (Please see questions below for specifics).
2. Many technical details are missing.
3. Some conclusions are not thoroughly explained.
4. The paper is missing an impact statement

To summarize, while I think the contribution is important, I found too many missing details to make the contribution and claims sound, and some overloading of notation that made some parts difficult to follow. Therefore I am scoring a weak reject for now. However, if the authors answer the questions below and revise the paper accordingly, I would be open to updating my score.

Some minor issues that did not affect my score, but would improve the paper:
1. There are some typos and editing errors, such as line 123 right column: "Since method has since been".
2. Some papers in the references refer to the arxiv versions and not to the official proceedings publication, e.g. "Arjovsky, M. and Bottou, L. Towards Principled Methods for Training Generative Adversarial Networks, January 2017. URL http://arxiv.org/abs/1701.04862. arXiv:1701.04862 [stat]." should be "Arjovsky, M. and Bottou, L., 2017, February. Towards Principled Methods for Training Generative Adversarial Networks. In International Conference on Learning Representations."

---

> ### Author Rebuttal · Authors · 2026-03-31
>
> We sincerely appreciate the clarity of your comments as well as the opportunity to respond with the possibility of improving your score. We provide detailed responses to  Questions 1–9 below.
>
> - (1) Thank you for pointing this. It can indeed be shown that Precision and Recall (and, more generally, any point on the Precision–Recall curve defined in [1]) can be expressed as $1 - D_f(P \Vert Q)$ for a specific choice of $f$ [2]. Therefore, maximizing Precision and Recall is equivalent to minimizing the corresponding $f$-divergence. We will clarify this connection in the revised version. On the other hand, in Tables 1 & 2 we report the values of $\delta$-precision and $\delta$-recall, which we further discuss in our response in (9). These metrics measure the maximum difference between precision and recall across pairs of sensitive attributes. Hence, smaller values are better. The results in Tables 1 & 2 indicate that these differences are close to zero, which is a desirable outcome.
>
> - (2) For space reasons, we wanted Definition 2.2 to provides definitions for both $\delta$-EGO and $\delta$-MGO, which may lead to clarity issues. As rightly noted by the reviewer, the definition of $\delta$-EGO depends only on $Q$. It ensures that the proportions of the sensitive attributes in $Q$ are $\delta$-close (i.e., well balanced). The reason we introduce $P$ at the beginning of the definition is that it is required for the second part (second column of page 3), where we define $\delta$-MGO between $P$ and $Q$. We believe once everything is in order the definition is complete but will split it to improve readability.
>
> - (3+4) The goal of the experiments is to demonstrate that different models can have the same overall $f$-divergence, while exhibiting significantly different $f$-divergences at the level of sensitive attributes. The black curve represents the reference distribution. We define two subspace: $\mathcal{X}\_0 = \mathbb{R}\_{-}^{*}$ for $a=0$, and $\mathcal{X}\_1 = \mathbb{R}\_+$ for $a=1$. To illustrate our point, we consider 3 different models, shown using 3 shades of green (which we are happy to modify for clarity). Figures 1b, 1c, & 1d display the values of the divergence (both local & global), with red indicating higher values. Each model is represented by a marker. We observe that these three distributions share the same Jensen divergence (as shown by the level set in Figure 1b), while exhibiting strikingly different Jensen divergence values across sensitive attributes (see Figures 1c & 1d). Regarding the choice of the divergence value, we agree that selecting 1 hurts clarity. We have rerun the computations with a different $\epsilon$ obtained the same findings. We will include all details and the updated results in the revised version of the paper.
>
> - (5+6) Thank you very much for this remark. The correct sentence should read: "[...] set of distributions for which the conditional distributions with respect to $a$ belong to $\mathcal{Q}\_a$ for every $a \in \mathcal{A}$, and for which the proportions lie in [...]". The rest of the definition remains unchanged. The purpose of this definition is not to directly solve the problem, but rather to introduce the technical concepts required for the lower-bound proof in Theorem 4.3. This lower bound implies that, unless $\mathcal{Q} = \bar{\mathcal{Q}}\_{\mathcal{A}}$, the training scheme may not converge to the optimally fair model. Consequently, it is important to actively consider enforcing conditional constraints to approach this bound. We also acknowledge that the sentence "[...] promotes the minimization of the highest $f$-divergence [...]" may be misleading. What we intended to convey is that, if matching this lower bound, one would aim to minimize this value; however, this may not always hold in practice. We will clarify this point in the updated version of the paper.
>
> - (7) The advantage of using $f$-divergences is that most generative model objectives can be expressed as the minimization of such a divergence. In our case, both the MSE minimized for diffusion and the MLE used in next-token prediction can be related to the KL divergence. In this implementation, we compute a loss per class and then minimize the maximum across classes. More precisely, the selection of the class for which the loss is minimized is determined using an average over multiple gradient steps to improve stability. Our procedure is fully detailed in Algorithm 2 in Appendix B.3.2. We would be happy to include these details in an additional page, should the paper be accepted.
>
> - (8) We mostly present high-level intuitions mostly for lack of space, but we provide all details of in Appendix B. We will make an explicit mention of where to find these details and include key ones.
>
> - (9) Similar to Definition 4.1, the $\delta$-metrics measure the maximum difference of the metric computed across any pair of sensitive attributes. We will clarify this point in the revised version of the paper.

---

> > ### Author Rebuttal · Reviewer_vNZS · 2026-04-01
> >
> > The authors answered all my questions and clarified all misunderstandings, however in their answer to (1), they did not include the papers they cite [1,2].
> >
> > For Fig. 1, I would suggest using different colors and widths to all curves, changes the font colors of $a=0$, and $a=1$ because it can confuse with the blue and red of figures b-d, and either include a color legend, or preferably a description of the colors in the caption.
> >
> > I also agree with points 3-5 of reviewer GDKT, and think the paper would benefit from simplifying some of the definitions and numbering some of the equations.
> >
> > I am willing to increase my score to 4, but I first ask the authors to address the impact statement, include the missing citations from the rebuttal, and also include the updated text (as is possible under the 5000 characters constraint) with priority to (6), then (1) and then (7).

---

> > > ### Author Response · Authors · 2026-04-08
> > >
> > > Thank you very much for the follow up feedback. We thank the reviewer for the comment regarding Figure 1 and will revise it accordingly. We also agree with points 3–5 raised by Reviewer GDKT and will incorporate the suggested changes. We now provide the requested additionnal details below.
> > >
> > > **Impact statement:** This work introduces a definition of fairness tailored to generative models that explicitly enforces comparable generation quality across sensitive groups. By going beyond existing, less restrictive formulations, it provides a more principled framework for identifying and quantifying latent disparities in model behavior, with the goal of promoting more equitable systems.
> > > At the same time, we emphasize the inherent trade-offs associated with imposing such fairness constraints. In particular, improving parity across groups may lead to reductions in aggregate performance and a redistribution of errors toward groups that were previously better served. These effects underscore the need for careful, context-aware evaluation when deploying fairness-aware generative models in practice.
> > >
> > >
> > > **(6)** This result shows that, under a $\delta$-EGT constraint, it is not possible to obtain a model whose global $f$-divergence is smaller than the maximum divergence across group-conditional distributions. This highlights that, when designing models subject to EGT, particular attention should be paid to the worst-case (i.e., maximum) group-conditional divergence. Moreover, if the lower bound were tight, enforcing EGT could be interpreted directly as minimizing this maximum group-conditional divergence.
> > >
> > > **(1) Precision–Recall and $f$-divergence**
> > > Precision and Recall are widely used metrics for evaluating generative models [3, 4]. They admit a natural connection to $f$-divergences: under the generalized formulation proposed in [1], Precision and Recall can be expressed in terms of $f$-divergences [2] as
> > > $$
> > > \mathrm{Precision} = 1 - D_{f_{\mathrm{precision}}}(P \| Q), \quad
> > > \mathrm{Recall} = 1 - D_{f_{\mathrm{recall}}}(P \| Q),
> > > $$
> > > for appropriately chosen functions $f_{\mathrm{precision}}$ and $f_{\mathrm{recall}}$.
> > >
> > > Accordingly, we adopt Precision and Recall as primary evaluation metrics. When reported, both metrics are to be maximized, which is consistent with minimizing the corresponding $f$-divergences.
> > >
> > > [1] Sajjadi et al. https://arxiv.org/abs/1806.00035
> > >
> > > [2] Verine et al. https://arxiv.org/abs/2305.18910
> > >
> > > [3] Dhariwal et al. http://arxiv.org/abs/2105.05233
> > >
> > > [4] Song et al. http://arxiv.org/abs/2303.01469
> > >
> > >
> > > **(7) Min–max training** (page 7, after Methods and implementation details).
> > > We implement the min–max objective by leveraging the fact that standard generative training losses correspond to minimizing specific $f$-divergences. In particular, the mean squared error (MSE) used for diffusion models and the maximum likelihood objective used for next-token prediction both correspond to minimizing a Kullback–Leibler divergence [5]. Therefore, we do not compute an explicit $f$-divergence; instead, we use the model training loss as a proxy.
> > >
> > > At each iteration, given a minibatch with group labels $a \in \mathcal{A}$, we compute group-wise losses $ \{\ell_a \}$ , where $\ell_a$ is the empirical loss over samples belonging to group $a$. We maintain an exponential moving average (EMA) of these losses,
> > > $$
> > > L_a \leftarrow \alpha L_a + (1 - \alpha)\,\ell_a,
> > > $$
> > > to reduce variance in group selection. We then select the worst-performing group
> > > $$
> > > a^\star = \arg\max_{a \in \mathcal{A}} L_a,
> > > $$
> > > and update the model parameters using only the gradient $\nabla_\theta \ell_{a^\star}$. This yields a stochastic approximation of the min–max objective, where the maximization over groups is implemented via EMA-smoothed losses. The complete procedure is given in Algorithm 2 (Appendix B.3.2).
> > >
> > > [5] Song et al. http://arxiv.org/abs/2101.09258
> > >
> > > Finally, to address Reviewer GDKT’s concern regarding the generality of our claims, we have added two additional experiments on CIFAR and AFHQ-v2. These results further support and reinforce our findings across datasets. More details are provided in our response to Reviewer GDKT.

---

### Decision · Program_Chairs · 2026-04-30

**Decision:**

Accept (regular)

**Comment:**

The main idea of the work is to generalize ways of comparing (output) distributions for the purpose of evaluating fairness to allow for relative equality of "treatment" between groups. The authors show that this relative measure lends itself to a nice optimization formalism and present results along these lines.

Reviewers and authors had a good discussion that lead to score increases overall, and the paper didn't receive a full complement of reviews.